# Yellow sea mediated segregation between North East Asian *Dryophytes* species

**Amaël Borzée**[1]*, **Kevin R. Messenger**[1ʘ], **Shinhyeok Chae**[2ʘ], **Desiree Andersen**[3],
**Jordy Groffen**[3,4], **Ye Inn Kim**[3], **Junghwa An**[5], **Siti N. Othman**[3], **Kyongsin Ri**[6], **Tu
Yong Nam**[7], **Yoonhyuk Bae**[3], **Jin-Long Ren**[8], **Jia-Tang Li**[8], **Ming-Feng Chuang**[3],
**Yoonjung Yi**[3], **Yucheol Shin**[1], **Taejoon Kwon**[2], **Yikweon Jang**[3], **Mi-Sook Min**[9]*

1 College of Biology and the Environment, Nanjing Forestry University, Nanjing, Jiangsu, People's Republic of China, 2 Department of Biomedical Engineering, School of Life Sciences, Ulsan National Institute of Science and Technology (UNIST), Ulsan, Republic of Korea, 3 Department of Life Science and Division of EcoScience, Ewha Womans University, Seoul, Republic of Korea, 4 Department of Fish and Wildlife Conservation, Virginia Tech, Blacksburg, Virginia, United States of America, 5 National Institute for Biological Resources, Animal Resources Division, Incheon, Republic of Korea, 6 Department of International Economic Cooperation, Ministry of Land and Environment Protection, Pyongyang, Democratic People's Republic of Korea, 7 Institute of Zoology, State Academy of Science, Daesong-dong, Daesong District, Pyongyang, Democratic People's Republic of Korea, 8 Chengdu Institute of Biology, Chinese Academy of Sciences, Chengdu, People's Republic of China, 9 Research Institute for Veterinary Science, College of Veterinary Medicine, Seoul National University, Seoul, Republic of Korea

ʘ These authors contributed equally to this work.
* amaelborzee@gmail.com (AB); minbio@yahoo.co.kr (MSM)

**Data Availability Statement:** All relevant geographic data are within the manuscript and its Supporting Information files. Genetic data was submitted to the European Nucleotide Archive

## Abstract

While comparatively few amphibian species have been described on the North East Asian mainland in the last decades, several species have been the subject of taxonomical debates in relation to the Yellow sea. Here, we sampled *Dryophytes* sp. treefrogs from the Republic of Korea, the Democratic People's Republic of Korea and the People's Republic of China to clarify the status of this clade around the Yellow sea and determine the impact of sea level change on treefrogs' phylogenetic relationships. Based on genetics, call properties, adult morphology, tadpole morphology and niche modelling, we determined the segregated status species of *D. suweonensis* and *D. immaculatus*. We then proceeded to describe a new treefrog species, *D. flaviventris* sp. nov., from the central lowlands of the Republic of Korea. The new species is geographically segregated from *D. suweonensis* by the Chilgap mountain range and known to occur only in the area of Buyeo, Nonsan and Iksan in the Republic of Korea. While the Yellow sea is the principal element to the current isolation of the three clades, the paleorivers of the Yellow sea basin are likely to have been the major factor for the divergences within this clade. We recommend conducting rapid conservation assessments as these species are present on very narrow and declining ranges.

(ENA) under the accession number PRJEB36680
(https://www.ebi.ac.uk/ena/data/view/
PRJEB36680)

**Funding:** This work was supported by a
Conservation Research grant from The Biodiversity
Foundation to AB, a research grant from the
National Research Foundation of Korea
(2017R1A2B2003579) to YJ, and by a grant from
the National Institute of Biological Resources
(NIBR), funded by the Ministry of Environment
(MOE) of the Republic of Korea (NIBR201803101)
to MSM and TK.

**Competing interests:** The authors have declared
that no competing interests exist.

# Introduction

Sea level fluctuations in relation to climatic oscillations have consecutively isolated and connected populations [1–3]. In some cases, the interval between sea level variations resulted in drift and speciation [4], in others, the clades were brought back in contact before isolation [5]. For instance, Hylid treefrogs isolated on peninsulas in the Mediterranean sea diverged as a result of sea level variations in conjunction with ice ages [6, 7], while other species such as green toads (*Bufo viridis* subgroup) show widespread hybridisation in contact zones [8]. In addition, other species display "speciation reversal" following hybridisation and reversal into a shared gene-pool [9], such as sticklebacks [10].

Despite being one the of the largest seas in the world, the Yellow sea is a comparatively shallow water body [11] resulting from the submergence of the continental shelve [11]. That submergence is similar to that of most continental shelves since the Last Glacial Maximum [12–16] and comes as a result of climatic oscillations [13, 17]. The continental shelf on which the Yellow sea currently lies was totally exposed during the Last Glacial Maximum (hereafter LGM; 23–15.4 cal. kyr B.P; [11, 18]) with the water level at its minimum and about 130 m lower than today [13, 19–24]. At that period, the coastline had migrated about 1200 km seawards [18, 25, 26] and since then the water level rose to reach today's level [11, 27, 28]. In addition, the exposition of the continental shelf resulted in the creation of deltas by paleorivers, such as the Changjiang [29], Yangtzae and Yellow Rivers [30], providing large bodies of fresh water. Finally, most landscapes were free of ice during the quaternary ([2]; Qiu *et al.*, 2011) and the LGM [31, 32], allowing population movements.

Sea level oscillation had an impact on species present in the area of the Yellow sea and North East Asia in general [33–35], resulting in the absence of marine species during the low sea level period before recolonisation [36, 37]. Terrestrial species benefited from the low sea level and connective corridors [38, 39], but saw their range constricted or divided following the subsequent rise, reaching its current level about 7 kya [11]. For instance, species distributed in coastal areas on the Korean Peninsula are expected to have seen their range constricted (e.g. *Hynobius* sp.; [40]). The same pattern is expected for species distributed across the Yellow sea, such as *Bufo gargarizans* [41] and *Pelophylax nigromaculatus* [42]. Other species are expected to have seen their range split during past fluctuations of the sea level, such as *Pelophylax chosenicus* and *P. plancyi* [43], and *Dryophytes suweonensis* and *D. immaculatus* [44–47], the focal species of this study.

Eurasian Hylids originated from Central America [48], having dispersed through the Bering pass in two waves [44, 49–53]. The first dispersion wave towards Eurasia happened 28 to 23 mya [54] and is now represented by *Hyla* sp. [54, 55], while the second wave arrived in Eurasia between 18.9 and 18.1 mya [54] and is now represented by the genus *Dryophytes* [56], restricted to North East Asia [46]. Within Asian *Dryophytes*, the first divergence is dated between 14 mya [57] and 5.1 mya [44, 45]. This split resulted in the clades comprising *D. japonicus*, *D. suweonensis* and *D. immaculatus*.

The relationship between *D. immaculatus* and *D. suweonensis* is a point of contention [44–47], with synonymy recommended based on genetic information, despite the absence of other characters used to makes this decision. Borzée [46] noted that the data currently available is not enough to support the synonymy of the two species, and reviewed osteological data supporting differences between the two clades. Here, we present data on morphometrics of adults and tadpoles, call properties, phylogenetic relationships and ecological models for these two clades. We also describe a cryptic third clade distributed south of the range of *D. suweonensis* and first assigned to this species [58]. Finally, we connect the impact of the Yellow sea level variations on the relationship of these three species.

## Material and methods

### Field sampling

Data for the three clades was collected between 2016 and 2019 in the Republic of Korea (hereafter R Korea), the Democratic People's Republic of Korea (hereafter DPR Korea) and the People's Republic of China (hereafter China; Fig 1). Call properties and morphological differences between clades were not known before this study, and therefore individual sampled in China were assigned to *D. immaculatus*, based on range, and individuals sampled in R Korea and DPR Korea were assigned to either *D. suweonensis* or the new clade based on the genetic analyses. We aimed to obtain call recording, morphometrics and genetic samples for each individual collected. To do so, we first quietly waited for 5 min upon arrival at a site to locate calling males. We then recorded calling individuals with a linear PCM recorder (Tascam DR-40;

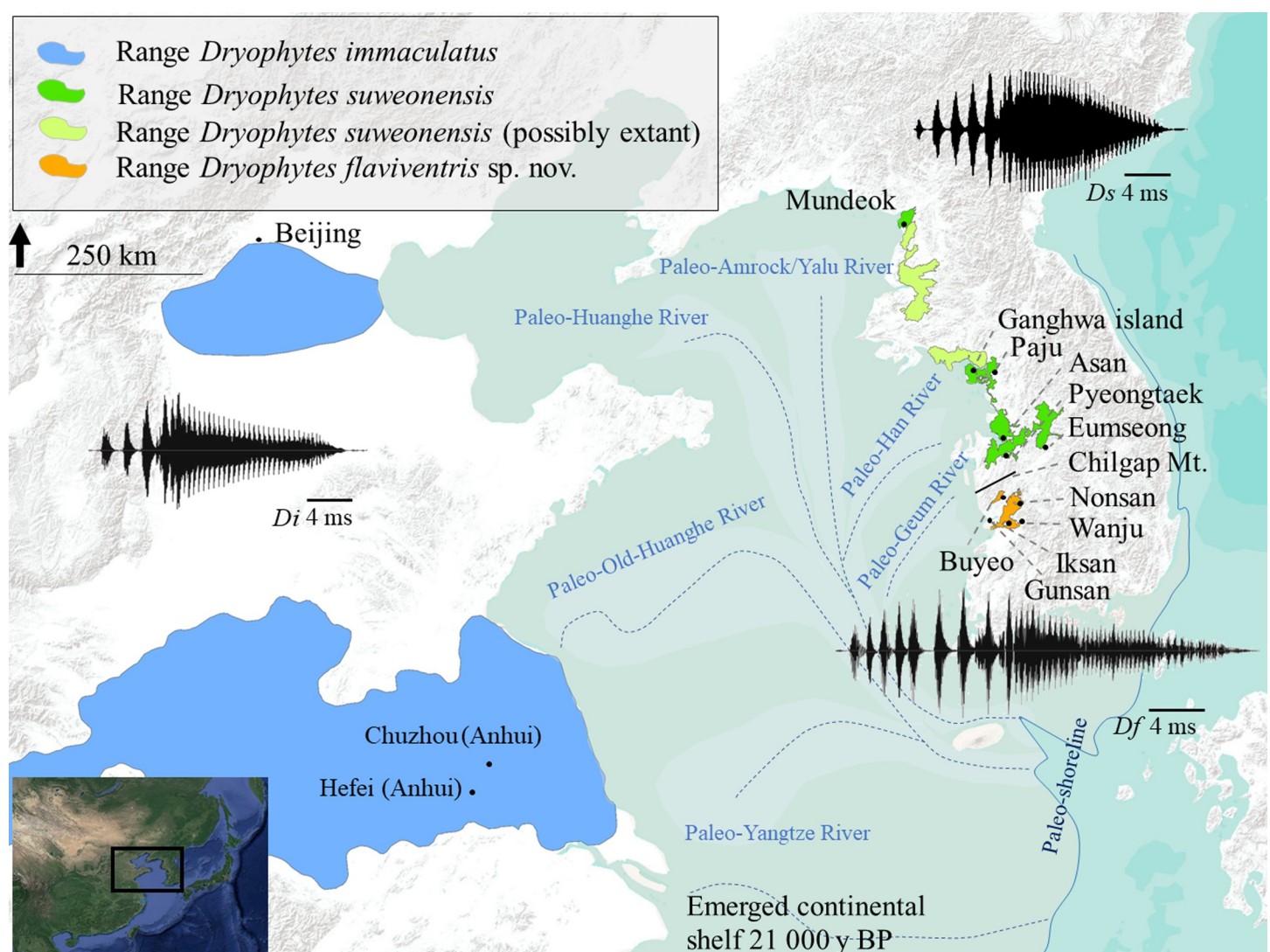

**Fig 1. Summary map including ranges, call properties and a phylogenetic tree including the three focal clades of this study: *Dryophytes suweonensis* and *D. flaviventris* sp. nov. and *D. immaculatus*.** Ranges are drawn based on [59–61] and the base layer was created in ArcMap 10.6 (desktop.arcgis.com; ESRI, Redlands, USA). Sampling localities are also included. The waveforms are not bound to axes but are shown to highlight the difference in the number of pulses in the three species. The dark-blue line is the sea shore 21,000 years BP (redrawn from [11]) and the dotted lines are paleorivers [62].

California, USA) linked to a unidirectional microphone (Unidirectional electret condenser microphone HT-81, HTDZ; Xi'an, China). Once recorded, the individual was caught and measured with a digital calliper (1108–150, Insize; USA) to the nearest 0.1 mm. Finally, each individual was orally swabbed to acquire genetic materials (cotton-tipped swab; 16H22, Medical Wire; Corsham, UK). Genetic materials were then stored at—20˚C until genetic analyses. Due to the cryptic nature of treefrogs, we did not manage to catch all frogs recorded, and not all frogs caught could be recorded. Sample sizes are explained below and summarized in Table 1.

The samples in Paju (R Korea) area were collected in 2013 under the Ministerial authorisation number 2013–16, while the other samples were collected in 2014 under the permits 2014–04, 2014–08 and 2014–20. Sampling in DPR Korea was conducted under the authorisation provided by the Ministry of Land and Environment Protection and sampling in PR China was conducted under the authorisation provided by Nanjing Forestry University. IACUC permits are not required when under ministerial authorisation for *D. suweonensis* and are not required for *D. immaculatus*.

## Genetic analyses: ddRAD-seq

The genetic analyses are based on genetic materials originating from four individuals sampled in Anhui (China; 32.310˚N, 118.583˚E), three individuals from Pyeongtaek (36.981˚N,

**Table 1. Sampling summary table.**

| (A) Samples size summary | | | |
|---|---|---|---|
| Species | Genetics | Calls | Morphometrics |
| *Dryophytes immaculatus* | 4 | 12 | 8 |
| *Dryophytes suweonensis* | 8 | 28 | 33 |
| *Dryophytes flaviventris* sp. nov. | 6 | 16 | 14 |
| (B) Origin samples for RAD-seq data (European Nucleotide Archive accession number PRJEB36680) | | | |
| Species | Country of origin | Locality | Voucher ID (alias) |
| *Dryophytes suweonensis* | Republic of Korea | Pyeongtaek | mms6883_HYLSU |
| *Dryophytes flaviventris* sp. nov. | Republic of Korea | Iksan | mms8551_HYLFL |
| *Dryophytes flaviventris* sp. nov. | Republic of Korea | Iksan | mms8552_HYLFL |
| *Dryophytes flaviventris* sp. nov. | Republic of Korea | Iksan | mms8553_HYLFL |
| *Dryophytes immaculatus* | People's Republic of China | Anhui | mms8665_HYLIM |
| *Dryophytes immaculatus* | People's Republic of China | Anhui | mms8670_HYLIM |
| *Dryophytes suweonensis* | Republic of Korea | Pyeongtaek | mms6884_HYLSU |
| *Dryophytes suweonensis* | Republic of Korea | Pyeongtaek | mms6885_HYLSU |
| *Dryophytes immaculatus* | People's Republic of China | Anhui | mms8666_HYLIM |
| *Dryophytes immaculatus* | People's Republic of China | Anhui | mms8667_HYLIM |
| *Dryophytes suweonensis* | Republic of Korea | Eumseong | mms4972_HYLSU |
| *Dryophytes suweonensis* | Republic of Korea | Eumseong | mms4973_HYLSU |
| *Dryophytes suweonensis* | Republic of Korea | Eumseong | mms4974_HYLSU |
| *Dryophytes suweonensis* | Republic of Korea | Eumseong | mms5027_HYLSU |
| *Dryophytes suweonensis* | Republic of Korea | Eumseong | mms5029_HYLSU |
| *Dryophytes flaviventris* sp. nov. | Republic of Korea | Iksan | mms8548_HYLFL |
| *Dryophytes flaviventris* sp. nov. | Republic of Korea | Iksan | mms8549_HYLFL |
| *Dryophytes flaviventris* sp. nov. | Republic of Korea | Iksan | mms8550_HYLFL |

The sample sizes for each clade used for the genetic analyses, call properties, and morphometrics are summarised here (A). The individuals for which DNA was extracted and RAD-seq data submitted to the European Nucleotide Archive (accession number PRJEB36680) are also listed in this table (B).

126.987˚E), five individuals from Eumseong (37.008˚N, 127.497˚E) and six individuals from Iksan (35.970˚N, 126.930˚E; R Korea; Fig 1). We extracted genomic DNA using a Quick-DNA Miniprep™ Plus Kit (Zymo #D4069), then 500 ng of genomic DNA was digested by 0.5 U of *BfaI* (NEB, Cat# R0568S) and *MboI* (NEB, Cat# R0147S) with 10X CutSmart buffer (NEB, Cat# B7204S) at 37˚C for 1 hour. We chose these enzymes because they showed the maximum coverage of DNA fragments with a length between 300 bp and 500 bp during preliminary analyses on *Xenopus tropicalis* genome (from XenBase http://www.xenbase.org/, RRID: SCR_003280). We selected and extracted all DNA fragments between 300 and 500 bp on a 1% agarose gel using the ZymoClean Gel DNA Recovery Kit (Zymo, Cat# D4008). The sequencing libraries were constructed with NEBNext® Ultra™ II DNA Library Prep Kit for Illumina® (NEB, Cat #7645) with 5–20 ng of size-selected DNA fragments, following the manufacturer's instructions. The final products in the range of 400 bp to 600 bp were cleaned on an E-gel CloneWell II Agarose Gels with SYBR Safe, 0.8% (ThermoFisher Cat# G661818).

We cleaned up the ddRAD-seq data with the 'process_radtags' program provided by STACKS (v 2.0; [63]). Because no reference genome was available, we used the "denovo_map.pl" program to build a population map containing sample information allowing for three mismatches between and within individuals. We further optimised the parameters of 'ustacks' to m = 4, M = 3 and n = 4, following the recommendation of the r80 method by Paris [64]. Next, we selected specific population reads for which more than 50% of loci were variable (-r 0.5). Finally, we averaged the $F_{ST}$ values among species using the "populations" program (Table 2).

To assess the population structure, we used the software STRUCTURE (v 2.3.4; [65, 66]). We ran the analysis with a 10,000 burn-in period and 50,000 MCMC repeats after the burn-in. To determine the best supported number of independent clades (K), we determined Delta K [67] through Structure-Selector [68] based on three iterations of STRUCTURE runs under the parameters described above.

To determine the relationship between the three clades, we used the output of the 'populations' program in STACKS. We then assigned each individual to one of the clades based on range, and constructed a phylogenetic tree using MrBayes (v 3.2.7a; [69]) before visualising the results in FigTree (v 1.4.4; [70]). We then inferred the phylogeny and the divergence time of three species groups using BEAST2 (v. 2.6.0; [71]), with 10,000,000 MCMC chain length and 10% burn-in.

## Call properties

For this analysis, we based our criteria on the work describing the call properties of *D. suweonensis* [72]. In this way, we were able to re-use 11 of the recordings to increase our sample size (total *n* = 56) and place our results in context provided by the literature on this clade. The recordings extracted from Park [72] originated from Asan (36.881˚N, 126.929˚E), Paju (37.899˚N, 126.764˚E) and Pyeongtaek (36.981˚N, 126.987˚E; R Korea; see Fig 1 for all localities). We recorded four individuals from Chuzhou (32.310˚N, 118.583˚E) and eight individuals from Hefei (32.310˚N, 118.583˚E), pooled under the label "Anhui" (China; *n* = 12); we also

**Table 2. Fixation index values ($F_{ST}$) values for *Dryophytes immaculatus, D. suweonensis* and *D. flaviventris* sp. nov. in North East Asia.**

| $F_{ST}$ | *D. suweonensis* | *D. flaviventris* sp. nov. |
|---|---|---|
| *D. immaculatus* | 0.101 | 0.100 |
| *D. suweonensis* | | 0.080 |

$F_{ST}$ was the highest between *D. immaculatus* and the other groups.

recorded seven individuals from Mundeok (39.471˚N, 125.386˚E; DPR Korea; *n* = 7), four individuals from Paju (37.899˚N, 126.764˚E; R Korea; combined with Park [72]; *n* = 6), three individuals from Asan (36.881˚N, 126.929˚E) and two individuals from Pyeongtaek (36.981˚ N, 126.987˚E; R Korea). We pooled together these last two populations and data from Park [72] under the name Pyeongtaek (*n* = 14) for subsequent analyses due to their origin from the same connected population [73]. Finally, we recorded seven individuals from Buyeo (36.244˚ N, 126.858˚E; R Korea; *n* = 7) and nine individuals from Iksan (35.970˚N, 126.930˚E; R Korea; *n* = 9). The population from Buyeo was assigned to the same clade as the one in Iksan based on landscape connectivity [73] as the closest population to the north is assigned to *D. suweonensis* through molecular tools but is segregated from the population in Buyeo by the mountainous range including Chilgap mountain (561 m a.s.l; Fig 1), a barrier to the species as it is not found above 120 m of altitude [74]. We also measured air temperature and relative humidity for each recording (Kestrel 5700; Boothwyn USA). The wording used to described calls followed that of Park ([72]; see Fig 2 therein), with advertisement calls composed of a train of notes, each of which consists of a series of pulses. We also used the same definitions to measure variables: note duration was the length of a note (s), although we also measured the duration of the connected pulse terminating a note (s) and determined the number of independent pulses present before the connected pulse. In addition, we measured peak dominant frequency (Hz) and 90% frequency bandwidth (Hz) for each of connected pulses and full notes. In total, we extracted data for 1166 notes.

Prior to data extraction, background noises were filtered out at 1 kHz, and spectrogram configuration was set at Hann window of 256-sample window size, 128-sample hop size with 50% frame overlap and 172- Hz frequency grid spacing. Each call was analysed for both temporal and spectral domains (Raven Pro 1.4; Cornell Lab of Ornithology, New York, USA), following the recommendations of Koehler [75]. To negate the effect of temperature on call properties, we adjusted the value of each variable to the average temperature of all recordings:

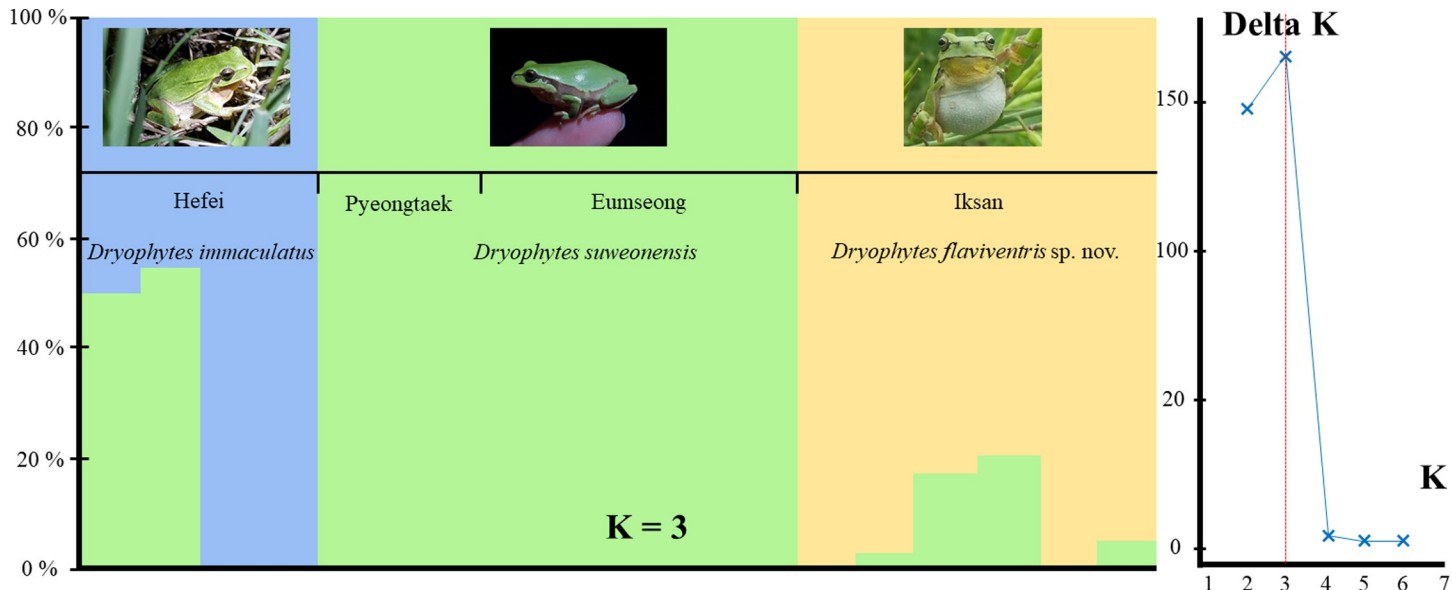

**Fig 2. STRUCTURE analysis of *Dryophytes immaculatus* (*n* = 4), *D. suweonensis* (*n* = 8) and *D. flaviventris* sp. nov. (*n* = 6) from North East Asia based on 8,949 ddRAD-seq loci.** The Delta K graph shows that K = 3 is supported, with hybrids detected in both *D. immaculatus* and *D. flaviventris* sp. nov. populations. Barplots show individual assignments. Names represent sampling localities (Fig 1) and waveforms are drawn to show correspondence with Figs 1 and 3.

21.69˚C. To do so, we computed the equation for the linear regression of each focal variable in function of temperature and calculated the adjusted variable at 21.69˚C.

As most extracted variables were correlated with each other (Pearson Correlation; $n = 1166$): duration of connected pulses ($0.22 < r2 < 0.77$, $p < 0.001$) and notes ($0.26 < r2 < 0.77$, $p < 0.001$), 90% frequency bandwidth for connected pulses ($0.09 < r2 < 0.92$, $p < 0.001$) and notes ($0.14 < r2 < 0.97$, $p < 1.001$), dominant frequency for connected pulses ($r2 = 0.06$, $p < 0.021$) and notes ($0.09 < r2 < 0.98$, $p < 0.002$) and the number of independent pulses ($0.06 < r2 < 0.54$, $p < 0.021$) we used a Principal Component Analysis to test for variations in call properties between populations. The PCA was based on 302 pulses from *D. immaculatus* (*n* individual = 12), 530 pulses from *D. suweonensis* (*n* individual = 28) and 333 pulses from the third clade (*n* individual = 16).

The Principal Component Analysis was set such as principal components were to be extracted if their eigenvalue > 1 under a varimax rotation. Variables were selected as loading into a PC if > 0.55 (Table 3). Once the PCs extracted, we tested for significant differences between clades through a General Linear Model. We used a univariate GLM with clades as dependent variable, the PCs as independent variable, and locality as covariate to test for the presence of significant difference between and within clade, and thus determine the integrity of each clade tested here. We then assessed the variation between clades two-by-two through a Tukey test. The analysis was run under a main-effects model and all assumptions were fulfilled: we did not detect any outlier when examining boxplots, the data was normally distributed (Shapiro-Wilk test; $p > 0.169$) and there was homogeneity of variances (Levene's test; $p > 0.001$). In addition, we tested for the presence of significant variation within each of the clades.

## Morphometrics adult frogs

The dataset for morphometrics was restricted compared to the one on call properties. We measured two frogs in Mundeok (39.471˚ N, 125.386˚ E; DPR Korea), four frogs in Asan (36.881˚

**Table 3. Principal components and their characteristic resulting from the PCA computed to segregate acoustic traits between the three *Dryophytes* clades sampled in PR China, DRP Korea and R Korea.**

| Principal components | PC1 | PC2 | Localities |
|---|---|---|---|
| Duration of connected pulses | - 0.15 | **0.85** | |
| 90% frequency bandwidth of connected pulses | **0.84** | 0.18 | |
| Peak dominant frequency of connected pulses | **0.65** | -0.13 | |
| Number of independent pulses | 0.28 | **0.67** | |
| Notes duration | 0.11 | **0.94** | |
| 90% frequency bandwidth of notes | **0.94** | 0.14 | |
| Peak dominant frequency of notes | **0.89** | 0.09 | |
| Eigenvalues | 3.112 | 1.91 | |
| % of Variance | 44.60 | 27.02 | |
| GLM | | | |
| $\chi2$ | 0.32 | 0.52 | 0.09 |
| Df, Df$_{error}$ | 2,1165 | 2,1165 | 2,1165 |
| *F* | 7.08 | 11.59 | 2.01 |
| *p* | **< 0.001** | **< 0.001** | 0.157 |

In bold are variables retained as loading into one if the PCs, and significant *p*-values from the statistical analysis to test for differences between clades. The PCA was based on 302 pulses from *D. immaculatus* (*n* = 12), 530 pulses from *D. suweonensis* (*n* = 28) and 333 pulses from *D. flaviventris* sp. nov. (*n* = 16).

N, 126.929˚E), two frogs in Pyeongtaek (36.981˚ N, 126.987˚E), four frogs in Paju (37.899˚N, 126.764˚E), five frogs in Buyeo (36.244˚N, 126.858˚E), seven frogs in Iksan (35.970˚N, 126.930˚E; R Korea) and eight frogs in Hefei (32.310˚N, 118.583˚E; China). In addition, we measured collection specimen preserved in alcohol. To ensure the absence of shrinking factor [76], we soaked each individual for five hours before measurement. We consider this procedure adequate as none of the alcohol-preserved samples presented either minimum or maximum value. From the collection specimen, we measured four individuals from Pyeongtaek (36.981˚ N, 126.987˚ E), eight individuals from Ganghwa Island (126.422˚ N, 37.646241, 126.422˚ E), seven individuals from Asan (36.881˚ N, 126.929˚ E), two individuals from Eumseong (37.008˚ N, 127.497˚ E) and two individuals from Iksan (35.970˚ N, 126.930˚ E). This resulted in a total of eight frogs for *D. immaculatus*, 33 frogs for *D. suweonensis* and 14 frogs for *D. flaviventris* sp. nov.

We measured morphological details known to be variable in this clade of Hylids [77]; SVL: snout-vent length; HLL: hind–limb length; MTW: toe webbing length between 2$^{nd}$ and 3$^{rd}$ toes; IND: inter-nostril distance, HW: head width, EL: eye length, EAD: distance between the anterior corners of the eyes, EPD: distance between the posterior corners of the eyes, EN: eye to nostril, NL: nostril to lip, HLt: head length, TD: tympanum diameter, dorsal patterns: presence of pattern on the back, pattern legs: presence of patterns (stripes) on the hind legs. Each variable was measured three times and averaged. Dorsal and leg patterns were discarded before analyses as they did not display any variations, as expected from the description of the holotype and the meaning of "*immaculatus*".

To be able to analyse morphometric variations without any bias due to the size of the individuals, we adjusted the dataset by dividing each value by the SVL of the individual. This procedure also eliminated any bias introduced by alcohol preserved samples. Variables in the dataset were generally correlated (Pearson's correlation; $n$ = 55; Table 4) so we used a PCA here as well to analyse variations between each of the clades. The Principal Component Analysis was set such that principal components were to be extracted if their eigenvalue > 1 under a varimax rotation. Variables were selected as loading into a PC if > 0.60 (Table 5). The PCA was based on eight *D. immaculatus* individuals, 33 *D. suweonensis* individuals and 14 individuals from *D. flaviventris* sp. nov.

Once the PCs were extracted, we tested for significant differences between clades through a General Linear Model. We used a univariate GLM with clades as dependent variable, the PCs as independent variable, and locality as covariate to test for the presence of significant difference between and within clades, and thus determine the integrity of each clade tested here. We then assessed the variation between clades two-by-two through a Tukey test. The analysis was run under a main-effects model and all assumptions were fulfilled: we did not detect any outlier when examining boxplots, the data was normally distributed (Shapiro-Wilk test; $p$ > 0.005) and there was homogeneity of variances (Levene's test; $p$ = 0.593). All biostatistical analyses were run in SPSS (SPSS, Inc., Chicago, USA).

### Tadpole morphology

Next, we looked at morphological variations between tadpoles of the three clades examined here, with a focus on oral morphology. However, we were unable to secure permits for tadpoles in Buyeo, Nonsan or Iksan and we were therefore unable to add data for *D. flaviventris* sp. nov. to this section of the manuscript. The data for *D. suweonensis* arises from five wild caught pairs with tadpoles reared in captivity (see [78] for details on rearing protocol) resulting in 100 tadpoles ($n$ = 20 per family) examined and photographed for their oral structure. The oral structure of *D. suweonensis* is the only Korean anuran for which it has yet to be described

**Table 4. Pearson correlation table for morphometric variables collected from eight *D. immaculatus* individuals, 33 *D. suweonensis* individuals and 14 *D. flaviventris* sp. nov.**

| | | HLL | MTW | IND | HW | EL | EAD | EPD | EN | NL | HLt | TD |
|---|---|---|---|---|---|---|---|---|---|---|---|---|
| Hind–limb length | R | | -0.42 | 0.25 | 0.06 | 0.23 | 0.11 | 0.19 | 0.33 | 0.51 | -0.01 | 0.58 |
| (HLL) | *p* | | **0.002** | 0.067 | 0.644 | 0.098 | 0.433 | 0.169 | **0.014** | **< 0.001** | 0.946 | **< 0.001** |
| Toe webbing length | | R | | -0.13 | 0.10 | -0.02 | 0.01 | 0.09 | -0.13 | -0.09 | 0.10 | -0.26 |
| (MTW) | | *p* | | 0.362 | 0.462 | 0.913 | 0.920 | 0.503 | 0.344 | 0.536 | 0.486 | 0.057 |
| Inter-nostril distance | | | R | | -0.09 | 0.39 | 0.28 | 0.52 | 0.38 | 0.42 | 0.07 | 0.43 |
| (IND) | | | *p* | | 0.501 | **0.003** | **0.038** | **< 0.001** | **0.004** | **0.001** | 0.608 | **0.001** |
| Head width | | | | R | | 0.17 | 0.10 | 0.24 | 0.05 | 0.13 | 0.27 | -0.03 |
| (HW) | | | | *p* | | 0.218 | 0.450 | 0.072 | 0.738 | 0.363 | **0.050** | 0.845 |
| Eye length | | | | | R | | 0.14 | 0.41 | 0.04 | 0.46 | 0.28 | 0.44 |
| (EL) | | | | | *p* | | 0.299 | **0.002** | 0.798 | **< 0.001** | **0.041** | **0.001** |
| Distance between the anterior corners of the eyes | | | | | | R | | 0.46 | 0.46 | 0.34 | 0.11 | 0.13 |
| (EAD) | | | | | | *p* | | **< 0.001** | **< 0.001** | **0.013** | 0.444 | 0.326 |
| Distance between the posterior corners of the eyes | | | | | | R | | | 0.46 | 0.48 | 0.21 | 0.51 |
| (EPD) | | | | | | *p* | | | **< 0.001** | **< 0.001** | 0.120 | **< 0.001** |
| Eye to nostril | | | | | | R | | | | 0.25 | 0.02 | 0.37 |
| (EN) | | | | | | *p* | | | | 0.071 | 0.861 | **0.006** |
| Nostril to lip | | | | | | R | | | | | 0.17 | 0.56 |
| (NL) | | | | | | *p* | | | | | 0.217 | **< 0.001** |
| Head length | | | | | | R | | | | | | 0.22 |
| (HLt) | | | | | | *p* | | | | | | 0.099 |
| Tympanum diameter | | | | | | R | | | | | | |
| (TD) | | | | | | *p* | | | | | | |

The data presented is adjusted for snout-vent-length to prevent bias due to size variation.

[79], although already illustrated [80]. The data for *D. immaculatus* is based on four tadpoles captured in Hefei (32.310˚N, 118.583˚ E) and observed under the microscope (Infinity1-1C, Lumenera Corporation; Nepean, Canada), and compared to the work presented in Fei [59]. For the comparison between the three clades, we focused on the number of upper and lower labial tooth rows and the presence of medial gaps.

## Ecological landscape preference

We then aimed at determining differences in landscape and terrain slope suitability between the three clades. To do so, we ran maximum entropy models using 19 bioclimatic variables interpolated to high resolution from long-term weather data and following the ANUCLIM scheme [81] and slope (Digital Elevation Model from USGS) at 0.04 decimal degree resolution on Maxent (v. 3.4.0; [82]). While some variables selected may have been correlated, the ecology of *D. immaculatus* is relatively unknown and some reports hint at a potential presence at higher altitudes (http://www.amphibiachina.org/species/307). In addition, baseline ecological niche modelling has not yet been conducted for these species and variables of interest are still questionable. Therefore, to avoid the exclusion of relevant variables [83], but also to create a baseline for future studies, we decided to include all bioclimatic variables and thus ensure the absence of preconceived bias on the ecological variables relevant to a species not yet described. In this framework, the inclusion of all variables further allowed us to compare the importance of each variables amongst species while a selection would have resulted in the use of different

**Table 5. Principal components and their characteristic resulting from the PCA computed to segregate morphological traits between the three *Dryophytes* clades sampled in PR China, DRP Korea and R Korea.**

| Principal components | 1 | 2 | 3 | 4 | Location |
|---|---|---|---|---|---|
| Hind–limb length | 0.30 | 0.15 | **0.81** | 0.07 | |
| Toe webbing length | 0.04 | 0.04 | **-0.80** | 0.14 | |
| Inter-nostril distance | **0.65** | 0.40 | 0.04 | -0.30 | |
| Head width | -0.06 | 0.15 | 0.03 | **0.88** | |
| Eye length | **0.81** | -0.07 | -0.01 | 0.20 | |
| Distance between the anterior corners of the eyes | 0.11 | **0.80** | -0.07 | 0.10 | |
| Distance between the posterior corners of the eyes | 0.58 | **0.61** | -0.08 | 0.18 | |
| Eye to nostril | 0.08 | 0.81 | 0.27 | -0.05 | |
| Nostril to lip | **0.66** | 0.27 | 0.30 | 0.15 | |
| Head length | 0.37 | -0.05 | -0.13 | **0.60** | |
| Tympanum diameter | **0.69** | 0.18 | 0.47 | 0.01 | |
| Eigenvalues | 3.69 | 1.61 | 1.29 | 1.02 | |
| % of Variance | 33.62 | 14.66 | 11.77 | 9.26 | |
| GLM | | | | | |
| $\chi 2$ | 4.64 | 0.47 | 14.71 | 1.29 | 0.03 |
| Df, $Df_{error}$ | 2,54 | 2,54 | 2,54 | 2,54 | 2,54 |
| F | 5.4 | 0.46 | 31.12 | 1.31 | 0.07 |
| *p* | **0.007** | 0.630 | **< 0.001** | 0.278 | 0.791 |

In bold are variables retained as loading into one if the PCs, loading if > 0.6. Significant *p*-values from the statistical analysis to test for differences between clades are in bold as well. The data was collected from eight *D. immaculatus* individuals, 33 *D. suweonensis* individuals and 14 *D. flaviventris* sp. nov.

best-fit models for each species and it would have therefore prevented us from comparing the response variables.

The models were run with data extracted from GBIF (DOI: 10.15468/dl.vdpjqu) combined to data collected by the authors (S1 Appendix table), ensuring that records were not duplicated, and removed when present. We used ten bootstrap replicates with a 20% random test percentage, and then used a jackknife test to determine the variables contributing the greatest amount to habitat suitability. The jackknife tests also served to elucidate the effects of individual environmental variables on the species' distributions. We then used a Tukey HSD test to compare permutation importance of variables between pairs and test for significant differences between clades.

Next, we tested for niche equivalency between the three clades, following the protocol of Warren [84], through the "nicheEquivalency" function from the "dismo package" [85]. We used 20 iterations for the package to make a null distribution by running iterations of maxent with randomized selection from pooled occurrence of two of the three clades, and then compared the results of these iterations to the actual niche overlap statistic. The analyses were run in ArcMap 10.6 (ESRI, Redlands, USA) and RStudio (RStudio Team, Integrated Development for R. RStudio, Inc., Boston, USA).

## Results

### Genetic analyses

To better understand the genetic variation between the different clades we performed a ddRAD-seq analysis based on about 10 million of paired-reads from each of the 18 individuals. We analysed the variations at 5,042 loci (SNPs = 5,819 and polymorphic loci = 2,561), present

in more than 50% of each group. The results showed that each of the three clades had a distinctive genetic structure compared to the other, seen through the phylogenetic tree (Fig 3) and the STRUCTURE plots (Fig 2)Fig. Based on the records of the TimeTree database [86], we estimated the divergence between *D. japonicus* and the *D. suweonensis* group c. 13.67 mya,. We inferred the divergence time between *D. suweonensis* and *D. immaculatus* to be c. 1.02 mya, followed by a divergence between *D. suweonensis* and *D. flaviventris* sp. nov. c. 0.97 mya (Fig 3). Regarding the STRUCTURE analysis, the optimal K was 3 based on the maximum Delta K analysis [67]. In addition, the fixation index ($F_{ST}$) was the highest between *D. immaculatus* and the other clades (Table 2).

## Call properties

In total, we extracted call property data for 302 notes for *D. immaculatus* (17.35 ± 12.22 per individual), 530 notes for *D. suweonensis* (11.72 ± 7.67 per individual) and 333 notes for *D. flaviventris* sp. nov. (16.19 ± 14.23 per individual). The PCA to identify the independent dimensions of the call properties between the three clades resulted in two Principal Components, with eigenvalues of 3.12 and 1.91, explaining a cumulated variation of 72.02% (Table 3). A variable was judged to be important if displaying a loading factor > 0.65, so that each variable loaded in one of the PCs. Based on the variables loading onto each of the PCs, we assigned PC1 to call frequency and PC2 to temporal properties (Table 3).

The GLM was significant for PC1 and PC2 (Table 3), highlighting significant differences in term of frequency and temporal properties between the three *Dryophytes* clades. The difference between localities was not significant ($p = 0.157$), highlighting the absence of significant variation within clades. When looking at the results of the Tukey tests to discriminate between

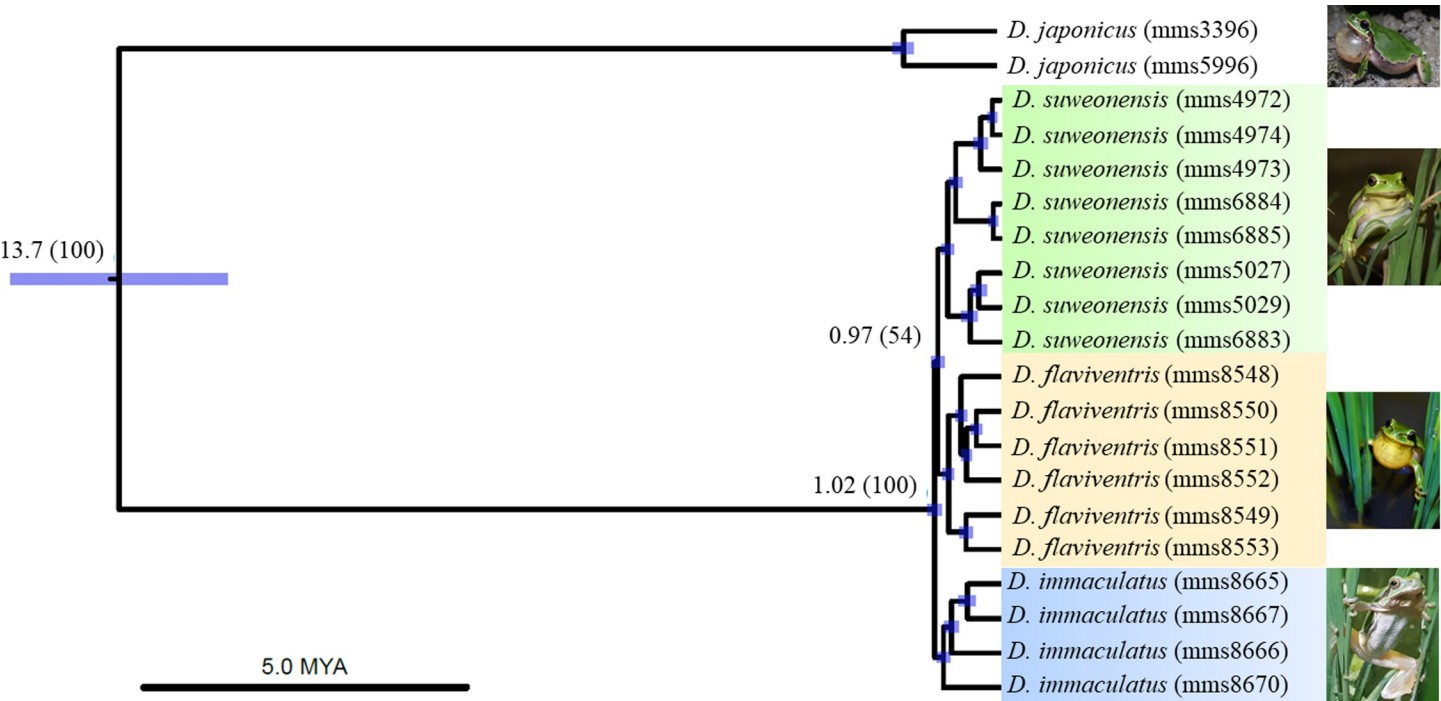

**Fig 3. Genetic structures of the *Dryophytes immaculatus* group.** Phylogenetic tree based on ddRAD-seq polymorphic loci, highlighting the divergence of *D. immaculatus* from the other clades 1.02 mya, and the split between *D. suweonensis* and *D. flaviventris* c. 0.97 mya. The estimated divergence time (in mya) is illustrated, together with a 95% confidence interval bar and the posterior probabilities (BEAST2) for each clade.

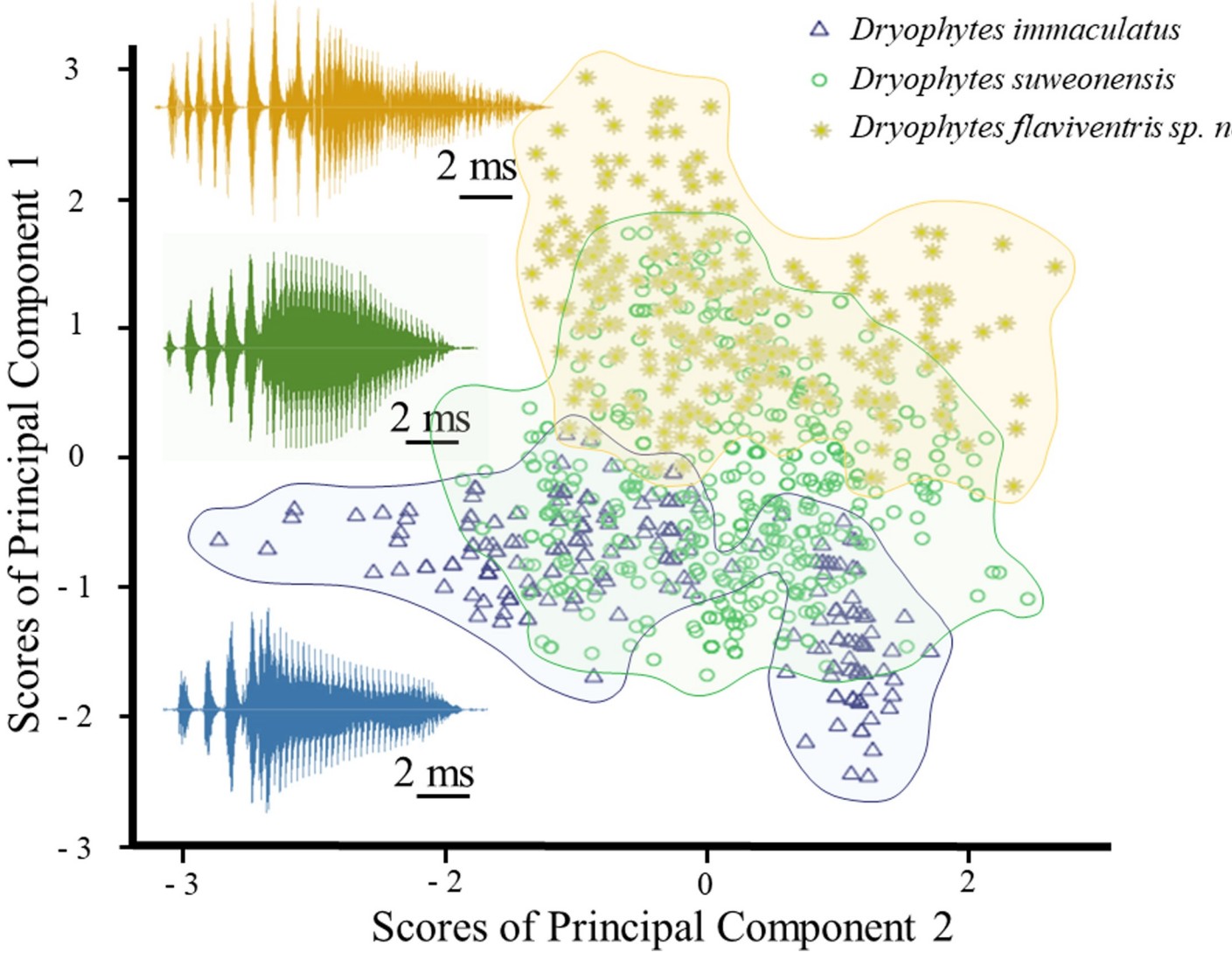

**Fig 4. Plot of all significant variables resulting from the PCA on the call properties of *Dryophytes immaculatus*, *D. suweonensis* and *D. flaviventris* sp. nov. from North East Asia.** The two variables are plotted against each other, highlighting a clustering of variables, although overlapping on extremes. The PCA was based on 302 pulses from *D. immaculatus* (*n* = 12), 530 pulses from *D. suweonensis* (*n* = 28) and 333 pulses from *D. flaviventris* sp. nov. (*n* = 16). The colour version is needed to match clouds of points and call properties.

clades, all three clades where significantly different from each other for temporal variables (p < 0.001). However, when focusing on variations in frequency, the difference between *D. suweonensis* and *D. flaviventris* sp. nov. did not reach significance (*p* = 0.192), while the difference between *D. suweonensis* and *D. immaculatus* was significant (*p* < 0.001). When plotting the two variables resulting from the PCA against each other a clustering pattern was apparent (Fig 4), although overlapping on minima and maxima.

Among the differences between the three clades (*n D. immaculatus* = 302, *n D. suweonensis* = 530, *n D flaviventris* sp. nov. = 333), an important distinction was the number of independent pulses before the connected pulse within each note, with the average number increasing on an East-West gradient around the Yellow sea (Table 6 and Fig 1).

**Table 6. Call properties for the three *Dryophytes* species present around the Yellow sea, when corrected for temperature variation (21.69˚C).**

| | | Mean | SD | Minimum | Maximum |
|---|---|---|---|---|---|
| Duration of connected pulses (s) | *D. immaculatus* | 0.082 | 0.01 | 0.06 | 0.09 |
| | *D. suweonensis* | 0.084 | 0.01 | 0.07 | 0.11 |
| | *D. flaviventris* sp. nov | 0.094 | 0.01 | 0.08 | 0.12 |
| 90% frequency bandwidth of connected pulses (Hz) | *D. immaculatus* | 2002.08 | 256.70 | 895.25 | 2464.68 |
| | *D. suweonensis* | 2139.15 | 218.20 | 1761.04 | 2889.28 |
| | *D. flaviventris* sp. nov | 2172.72 | 231.17 | 1877.63 | 3134.07 |
| Peak dominant frequency of connected pulses (Hz) | *D. immaculatus* | 3067.44 | 187.97 | 2848.32 | 3456.44 |
| | *D. suweonensis* | 3279.51 | 239.88 | 2861.80 | 3723.10 |
| | *D. flaviventris* sp. nov | 3207.28 | 143.95 | 2953.12 | 3542.95 |
| Number of independent pulses | *D. immaculatus* | 3.50 | 0.71 | 2.00 | 5.00 |
| | *D. suweonensis* | 5.34 | 0.80 | 4.00 | 7.00 |
| | *D. flaviventris* sp. nov | 6.64 | 0.73 | 5.00 | 8.00 |
| Notes duration (s) | *D. immaculatus* | 0.11 | 0.01 | 0.09 | 0.13 |
| | *D. suweonensis* | 0.12 | 0.02 | 0.10 | 0.17 |
| | *D. flaviventris* sp. nov | 0.15 | 0.01 | 0.13 | 0.19 |
| 90% frequency bandwidth of notes (Hz) | *D. immaculatus* | 2006.89 | 221.02 | 1402.33 | 2449.50 |
| | *D. suweonensis* | 2131.41 | 187.86 | 1673.83 | 2566.46 |
| | *D. flaviventris* sp. nov | 2195.16 | 257.78 | 1882.58 | 2631.47 |
| Peak dominant frequency of notes (Hz) | *D. immaculatus* | 2099.28 | 246.84 | 1305.75 | 2567.49 |
| | *D. suweonensis* | 2072.38 | 197.62 | 1607.36 | 2640.86 |
| | *D. flaviventris* sp. nov | 2199.86 | 260.55 | 1671.92 | 2651.68 |

Data based on 302 pulses from *D. immaculatus* (*n* = 12), 530 pulses from *D. suweonensis* (*n* = 28) and 333 pulses from *D. flaviventris* sp. nov. (*n* = 16).

## Morphometrics adult frogs

The PCA used to discriminate the morphological variations between the three clades resulted in four PCs (Table 5), explaining 69.31% of the variance. While most averages were different from each other (Table 7), only two of the PCs were significantly significant under the GLM (Table 5). Namely PC1, related to head structure and PC3 related to limbs. We did not detect any significant variation between populations within a clade (Table 5). When looking at the variations between each of the clade through the Tukey test, all three clades were different from each other for the PC3 ($p < 0.001$ for all comparisons, related to limbs). However, *D. immaculatus* and *D. suweonensis* were not significantly different for PC2 ($p = 0.774$), while *D. immaculatus* was significantly different from *D. flaviventris* sp. nov. ($p = 0.036$) and *D. suweonensis* was significantly different from *D. flaviventris* sp. nov. ($p = 0.003$). The segregation between the three clades was clear when PC1 and PC3 were plotted against each other, and none of the clades overlapped with each other (Fig 5). When looking at variables separately, a clear segregation between clades was the length of the webbing between 2nd and 3rd toes: almost absent in *D. immaculatus*, intermediate in *D. flaviventris* sp. nov. and the longest in *D. suweonensis* (Fig 6).

## Morphometrics tadpoles

We described the oral structure of *D. suweonensis* for tadpoles at Gosner stage > 26 and < 36 ([87]; Fig 7) such as non-emarginated 2(2)/3, meaning two anterior tooth rows with a median gap on the second anterior row and three posterior rows. This is also the pattern visible on the pictures of Kim [80], and also matches with the labial tooth rows formula of *D. japonicus* [79].

**Table 7. Descriptive statistics for three *Dryophytes* species from North East Asia.**

|     |     | Mean | SD | Minimum | Maximum | Range |
| --- | --- | --- | --- | --- | --- | --- |
| HLL | *D immaculatus* | 2.33 | 0.09 | 2.17 | 2.49 | 0.31 |
|     | *D suweonensis* | 1.40 | 0.12 | 1.11 | 1.65 | 0.54 |
|     | *D. flaviventris* sp. nov. | 2.03 | 0.43 | 1.21 | 2.45 | 1.23 |
| MTW | *D immaculatus* | 0.00 | 0.00 | 0.00 | 0.01 | 0.01 |
|     | *D suweonensis* | 0.12 | 0.11 | 0.07 | 0.76 | 0.69 |
|     | *D. flaviventris* sp. nov. | 0.07 | 0.04 | 0.02 | 0.17 | 0.15 |
| IND | *D immaculatus* | 0.09 | 0.00 | 0.08 | 0.10 | 0.01 |
|     | *D suweonensis* | 0.08 | 0.01 | 0.05 | 0.11 | 0.06 |
|     | *D. flaviventris* sp. nov. | 0.09 | 0.01 | 0.08 | 0.13 | 0.05 |
| HW | *D immaculatus* | 0.32 | 0.01 | 0.31 | 0.35 | 0.04 |
|     | *D suweonensis* | 0.34 | 0.03 | 0.31 | 0.52 | 0.21 |
|     | *D. flaviventris* sp. nov. | 0.33 | 0.03 | 0.29 | 0.38 | 0.09 |
| EL | *D immaculatus* | 0.11 | 0.01 | 0.09 | 0.12 | 0.03 |
|     | *D suweonensis* | 0.11 | 0.02 | 0.08 | 0.15 | 0.07 |
|     | *D. flaviventris* sp. nov. | 0.12 | 0.02 | 0.09 | 0.15 | 0.06 |
| EAD | *D immaculatus* | 0.19 | 0.01 | 0.18 | 0.21 | 0.03 |
|     | *D suweonensis* | 0.19 | 0.01 | 0.15 | 0.23 | 0.08 |
|     | *D. flaviventris* sp. nov. | 0.19 | 0.01 | 0.16 | 0.21 | 0.05 |
| EPD | *D immaculatus* | 0.29 | 0.01 | 0.28 | 0.31 | 0.03 |
|     | *D suweonensis* | 0.30 | 0.01 | 0.27 | 0.32 | 0.05 |
|     | *D. flaviventris* sp. nov. | 0.31 | 0.03 | 0.26 | 0.37 | 0.11 |
| EN | *D immaculatus* | 0.09 | 0.01 | 0.08 | 0.10 | 0.02 |
|     | *D suweonensis* | 0.08 | 0.01 | 0.06 | 0.11 | 0.05 |
|     | *D. flaviventris* sp. nov. | 0.08 | 0.01 | 0.06 | 0.11 | 0.06 |
| NL | *D immaculatus* | 0.08 | 0.00 | 0.07 | 0.09 | 0.01 |
|     | *D suweonensis* | 0.07 | 0.01 | 0.05 | 0.09 | 0.04 |
|     | *D. flaviventris* sp. nov. | 0.08 | 0.02 | 0.05 | 0.12 | 0.06 |
| HLt | *D immaculatus* | 0.24 | 0.02 | 0.21 | 0.26 | 0.05 |
|     | *D suweonensis* | 0.30 | 0.02 | 0.24 | 0.35 | 0.11 |
|     | *D. flaviventris* sp. nov. | 0.30 | 0.03 | 0.23 | 0.36 | 0.13 |
| TD | *D immaculatus* | 0.07 | 0.01 | 0.06 | 0.07 | 0.02 |
|     | *D suweonensis* | 0.06 | 0.00 | 0.05 | 0.07 | 0.02 |
|     | *D. flaviventris* sp. nov. | 0.07 | 0.01 | 0.05 | 0.09 | 0.04 |

The data presented is adjusted for snout-vent-length to prevent bias due to size variation and is based on eight *D. immaculatus* individuals, 33 *D. suweonensis* individuals and 14 *D. flaviventris* sp. nov.

The oral structure of *D. immaculatus* for tadpoles at Gosner stage > 26 and < 36 ([87]; Fig 7) was also non-emarginated, with a labial tooth row formula of 1/3(1), meaning a single anterior tooth row and three posterior tooth rows, with a median gap on the first posterior row. While Fei [59] does not provide the labial tooth row formula of *D. immaculatus*, the labial tooth row formula of all other Hylids is drawn such as 2(2)/3.

## Ecological landscape preference

The results of the MaxEnt models showed geographically different regions for peak habitat suitability for the three clades. For *D. immaculatus* (AUC = 0.981), the area with the highest habitat suitability is along the Yangtze river, and on the southern tip of the Korean Peninsula,

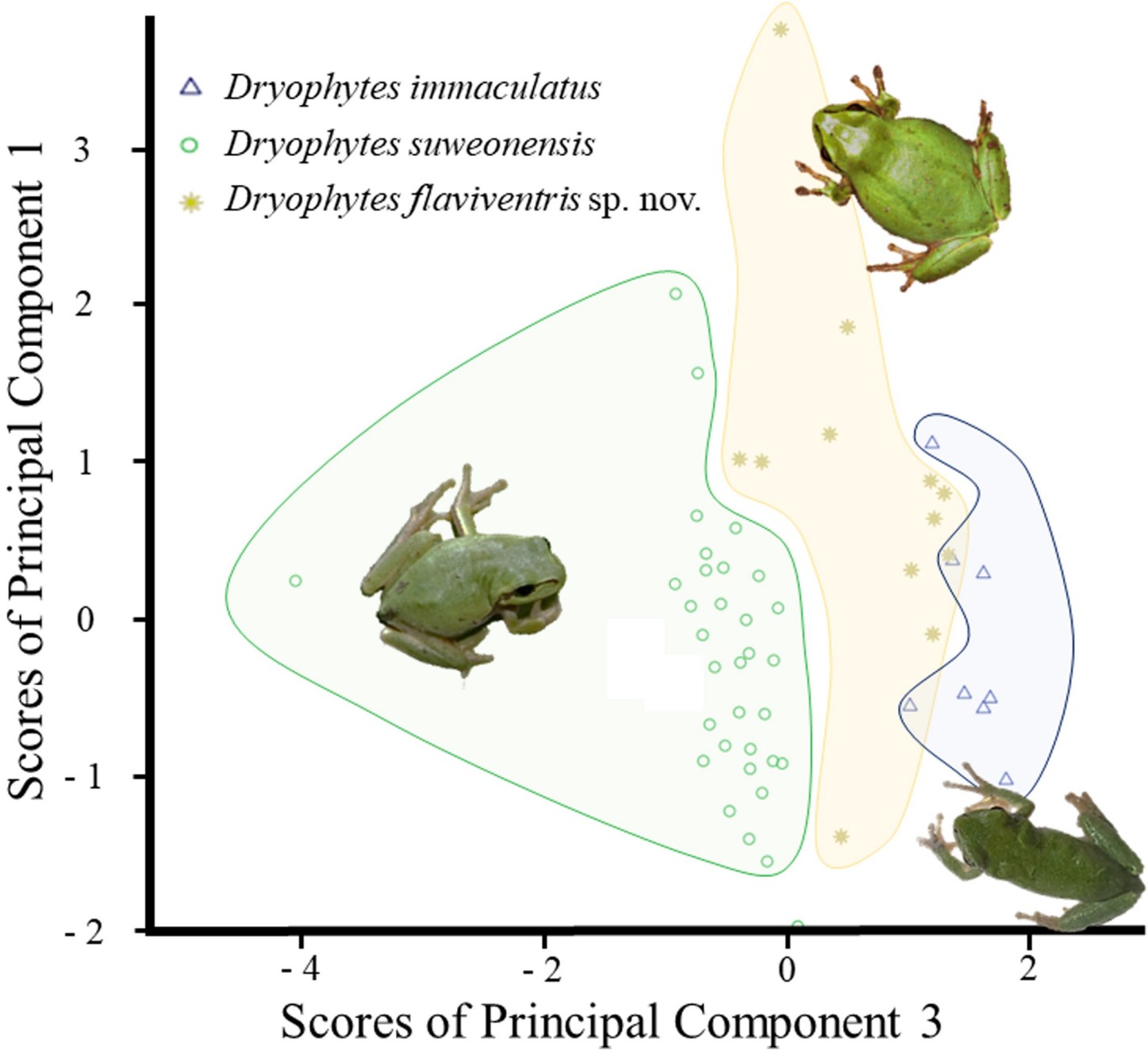

**Fig 5. Plot of the two significant variables resulting from the PCA on the morphology of *Dryophytes immaculatus*, *D. suweonensis* and *D. flaviventris* so. nov. from North East Asia.** The segregation between the three clades is clear when PC1 and PC2 are plotted against each other and none of the variable representative of the species overlaps with another species. The PCA was based on eight *D. immaculatus* individuals, 33 *D. suweonensis* individuals and 14 *D. flaviventris* sp. nov.

but further south than that of the two other clades (Fig 8). The area of maximum habitat suitability for *D. suweonensis* (AUC = 0.996) and *D. flaviventris* sp. nov. (AUC = 0.999) are geographically closer, but only weakly overlapping, and they are segregated around the area of Chilgap Mountain (Figs 1 and 7).

The Jackknife test to determine the variables of importance showed that most variables were important for at least one species. The Tukey HSD permutation tests to compare the

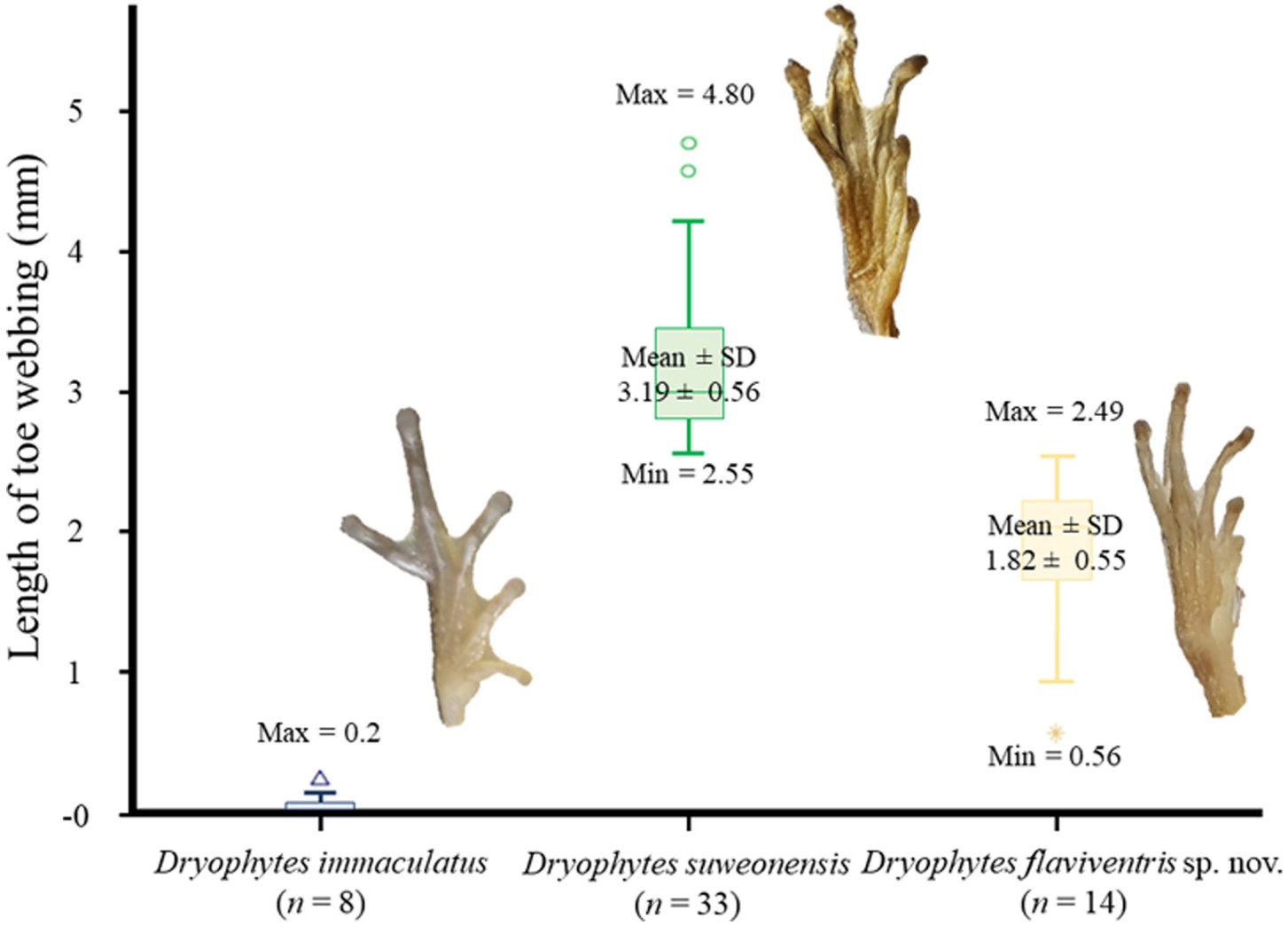

**Fig 6. Details of toe webbing for *Dryophytes immaculatus* (*n* = 8), *D. suweonensis* (*n* = 33) and *D. flaviventris* sp. nov. (*n* = 14).** Note the absence of overlap and the quasi absence of webbing between the 2nd and 3rd toes in *D. immaculatus*, in contrast with the two other species.

importance of variables between pairs and test for differences between clades highlighted significant differences between 27 clade pairs (Table 8). However, only the minimum temperature of coldest month (Bio6) was significantly different between all three species pair combinations. The difference between the clades for Bio6 and slope (Fig 9) shows that *D. suweonensis* is the clade best adapted to lowest minimum temperature of coldest month, while *D. immaculatus* is the species the least adapted to cold winters. On the other hand, *D. immaculatus* can live in landscapes with a slope significantly steeper than that of the two other clades (Fig 9).

The results of the niche equivalency showed a significant difference ($p < 0.0001$) between each clade in a two by two comparison: *D. immaculatus–D. suweonensis*: D = 0.147, I = 0.417; *D. immaculatus–D. flaviventris* sp. nov.: D = 0.140, I = 0.354; *D. suweonensis–D. flaviventris* sp. nov.: D = 0.425, I = 0.714). These results of the niche equivalency tests show that the overlap between clades is less than would be expected by random chance, wherein the absence of difference would assume that two populations are the same species with the same ecological requirements.

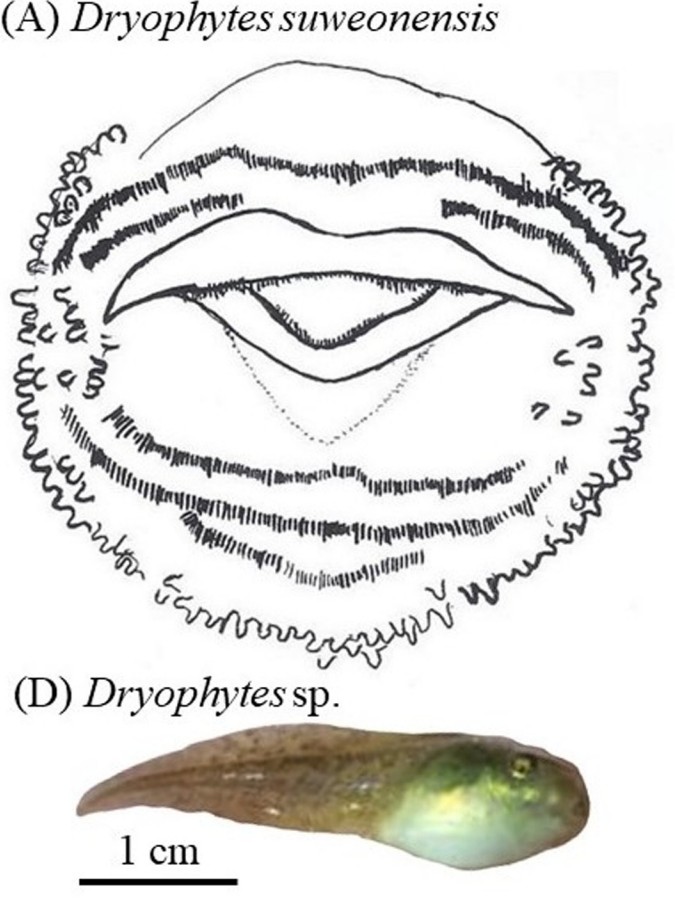

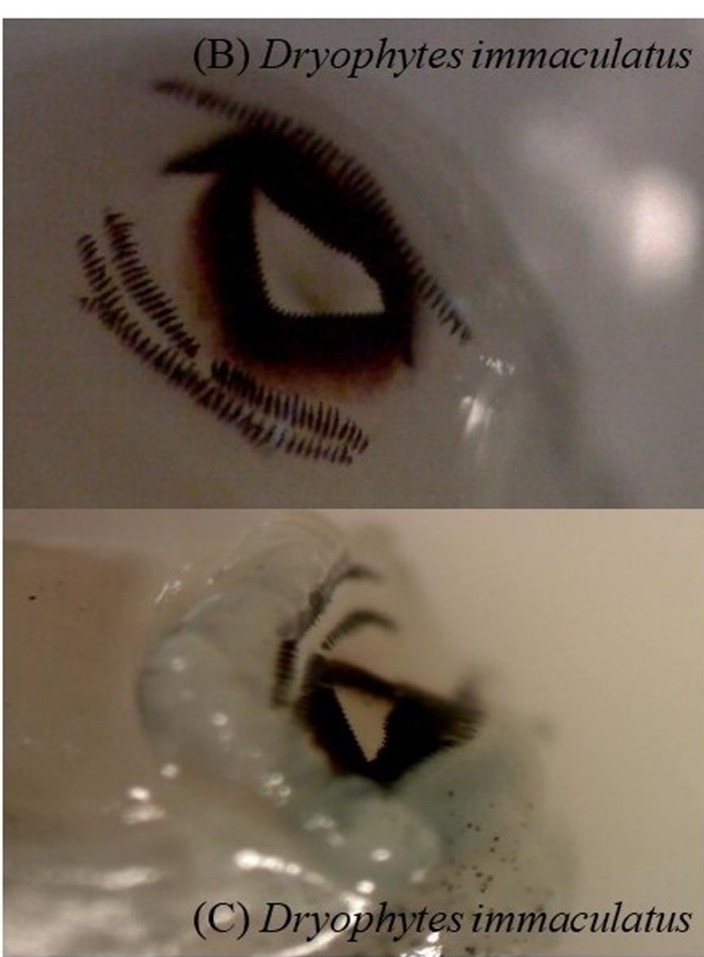

**Fig 7. Details of labial tooth row formula for *Dryophytes suweonensis* tadpoles (*n* = 100, A summarised in drawing; 2(2)/3) and *D. immaculatus* tadpoles (B, C; 1/3 (1); *n* = 4).** The labial tooth row formula structure for *D. immaculatus* deviates from other described Hylids from North East Asia. (D) is a tadpole from the range of *D. suweonensis* and may be an adequate representation of tadpoles for the whole group.

## Species description

Based on the combination of all variables described above, we formally describe a new species:

### *Dryophytes flaviventris* sp. nov. Borzée and Min, 2019

**Identity, distribution and diagnosis.** The first documented report of *D. flaviventris* was in 2016 [58], described as a range extension of *D. suweonensis* [60, 88, 89]. Populations are now known from Buyeo, Nonsan and Iksan (R Korea; Fig 1). Two contiguous populations were also known in Gunsan and Wanju, respectively extirpated putatively due to invasive American bullfrogs [90] and land conversion [60]. The species is not known to occur in any natural habitat following the conversion of all wetlands into rice paddies [91].

The species differs from *D. suweonensis* by slightly more elongated body and limbs, and an intermediate length of webbing between the 2nd and 3rd toes compared to *D. immaculatus* and *D. suweonensis* (Fig 6). The skeleton of the species has been described in great details by Kim [92], although assigned to *D. suweonensis*, and the skeleton of the digits was found to be significantly different from that of *D. immaculatus* ([46]; Table 2 therein). The species' pupils are horizontal and the skin texture differs ventrally (shagreened) and dorsally (smooth). No skin

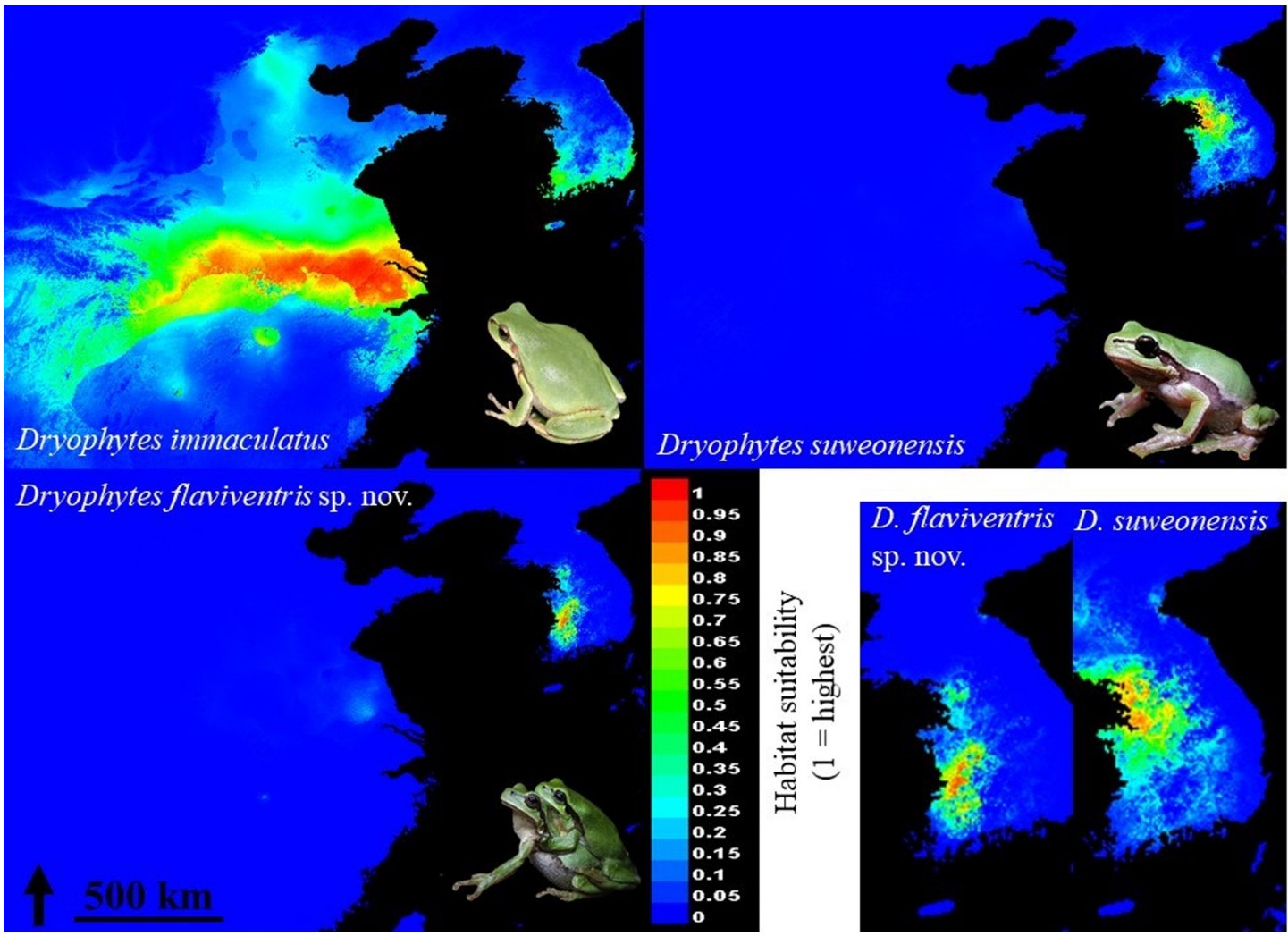

**Fig 8. Habitat suitability models for *Dryophytes immaculatus*, *D. suweonensis* and *D. flaviventris* sp. nov.** Maps were computed using Maxent (version 3.4.0; [82]) in ArcMap 10.6 (desktop.arcgis.com; ESRI, Redlands, USA) based on data originating from GBIF (DOI: 10.15468/dl.vdpjqu) and author's data (S1 Appendix). We used environmental layers consisting of 19 bioclimatic variables interpolated to high resolution from long-term weather data following the ANUCLIM scheme (worldclim.org; [85]) and slope (Digital Elevation Model from USGS) at 0.04 decimal degree resolution. Note the absence of overlap for habitat suitability > 0.6 for all three species.

folds are present and limbs are long and slender. Toes are rounded with circummarginal disks at their tips. Webbings between toes on the front legs are missing and vestigial between toes of the hind legs. No nuptial pads are visible. *Dryophytes flaviventris* is scansorial. The dorsum of *D. flaviventris* is light green generally without patches during the breeding season but it can also be dark grey or brown outside of the breeding season. The species is white with a general yellow hue on its immaculate ventrum, and a possible yellow lining below the black lateral line. The lateral line fades out half-way through the belly and inguinal loops have not been found on this species (Fig 10). The throat of males is yellow during the breeding season and it has not been recorded to be green–as seen in *D. suweonensis* in rare cases. Tadpoles of *D. flaviventris* have not been scientifically described but are expected not to be different from that of *D. suweonensis*, up to 3 cm long before limb emergence, with pointy tails, dorsolateral eyes and dorsal nostrils. Some tadpoles were found to have a reddish ventral coloration while the

**Table 8. Tukey HSD permutation tests on the three clades from the "*D. immaculatus* group".**

|        | Ds vs Di  | Ds vs Di | Df sp. nov. vs Di | Df sp. nov. vs Di | Df sp. nov. vs Ds | Df sp. nov. vs Ds |
|--------|-----------|----------|-------------------|-------------------|-------------------|-------------------|
| Bio1   | **0.008** | 8.64     | **< 0.001**       | 17.84             | **0.005**         | 9.21              |
| Bio2   | 0.276     | -0.53    | 0.115             | -0.70             | 0.870             | -0.17             |
| Bio3   | 0.786     | -0.32    | 0.052             | -1.20             | 0.190             | -0.88             |
| Bio4   | **0.001** | -10.00   | **0.001**         | -9.31             | 0.954             | 0.68              |
| Bio5   | **< 0.001** | 29.88  | 0.590             | -2.98             | **< 0.001**       | -32.86            |
| Bio6   | **0.002** | -27.64   | **0.013**         | 21.81             | **< 0.001**       | 49.45             |
| Bio7   | 0.054     | 0.57     | 1.000             | 0.00              | 0.054             | -0.57             |
| Bio8   | **0.023** | -9.68    | 0.533             | 3.71              | **0.002**         | 13.39             |
| Bio9   | **< 0.001** | 38.90  | 0.819             | 2.06              | **< 0.001**       | -36.85            |
| Bio10  | 0.533     | 0.73     | **0.041**         | -1.73             | **0.003**         | -2.46             |
| Bio11  | **0.003** | -24.98   | **0.003**         | -24.98            | 1.000             | 0.00              |
| Bio12  | 0.236     | -1.70    | 0.370             | -1.40             | 0.953             | 0.30              |
| Bio13  | 0.084     | -2.60    | 0.064             | -2.76             | 0.990             | -0.16             |
| Bio14  | **0.014** | -3.28    | 0.818             | 0.65              | **0.003**         | 3.93              |
| Bio15  | **0.004** | -6.72    | **0.004**         | -6.67             | 1.000             | 0.05              |
| Bio16  | 0.265     | -0.22    | 0.265             | -0.22             | 1.000             | 0.00              |
| Bio17  | **< 0.001** | 11.85  | 0.091             | 1.48              | **< 0.001**       | -10.36            |
| Bio18  | 0.318     | -1.72    | 0.721             | 0.90              | 0.081             | 2.62              |
| Bio19  | **0.016** | -0.80    | **0.016**         | -0.80             | 1.000             | 0.00              |
| Slope  | 0.975     | -0.37    | 0.054             | 4.30              | **0.034**         | 4.67              |

The tests were carried to compare the importance of variables and test for significant differences between clades for *D. immaculatus*, *D. suweonensis* and *D. flaviventris* sp. nov. in North East Asia. Ds = *Dryophytes suweonensis*; Di = *Dryophytes immaculatus*; Df = *D. flaviventris* sp. nov.

background colour is generally brown with darker patches. Fins are transparent and can display irregular dark patterns.

The behaviour of *D. flaviventris*, *D. immaculatus* and *D. suweonensis* when producing advertisement calls is similar, with males holding on the vegetation with their forelimbs (Fig 1). This behaviour is a key identification point segregating the *D. immaculatus* group from the *D. japonicus* group [93]. The advertisement calls of *D. flaviventris* have a similar pattern and structure as that of *D. immaculatus* and *D. suweonensis*, but a longer note length and a higher number of independent pulses before the connected pulse (Fig 2). Male *D. flaviventris* produce advertisement calls from mid-day, differing from *D. suweonensis* which starts calling in the mid-afternoon. Similarly to *D. suweonensis*, *D. flaviventris* is generally found on the vegetation at the edge of rice paddies outside of the calling period, and it hibernates in the vicinity of rice paddies [94]. The breeding season is related to rice paddy cultivation [95] and occurs between late April and early July. Males usually produce calls while hanging on vegetation with all four limbs (Fig 10). The species is declining and its range is constricting, principally because of habitat loss, and despite population dynamics being relatively better than that of *D. suweonensis* [96].

**Holotype.** Vouchers cgrb15733 (mms8555), adult male collected by Mi-Sook Min on 17 June 2018 in Iksan, Republic of Korea (35.984462 N, 126.921637 E; Fig 10), subsequently deposited at the Conservation Genetic Resource Bank, Seoul National University, Republic of Korea. The morphological measurements for the holotype are such as (in cm, measured three times and averaged): SVL: 28.59; MTW 2.94; IND 2.65; HW 9.74; EL 3.47; EAD 5.64; EPD 8.68; EN 2.87; NL 2.49; HLt 10.17; TD 2.10.

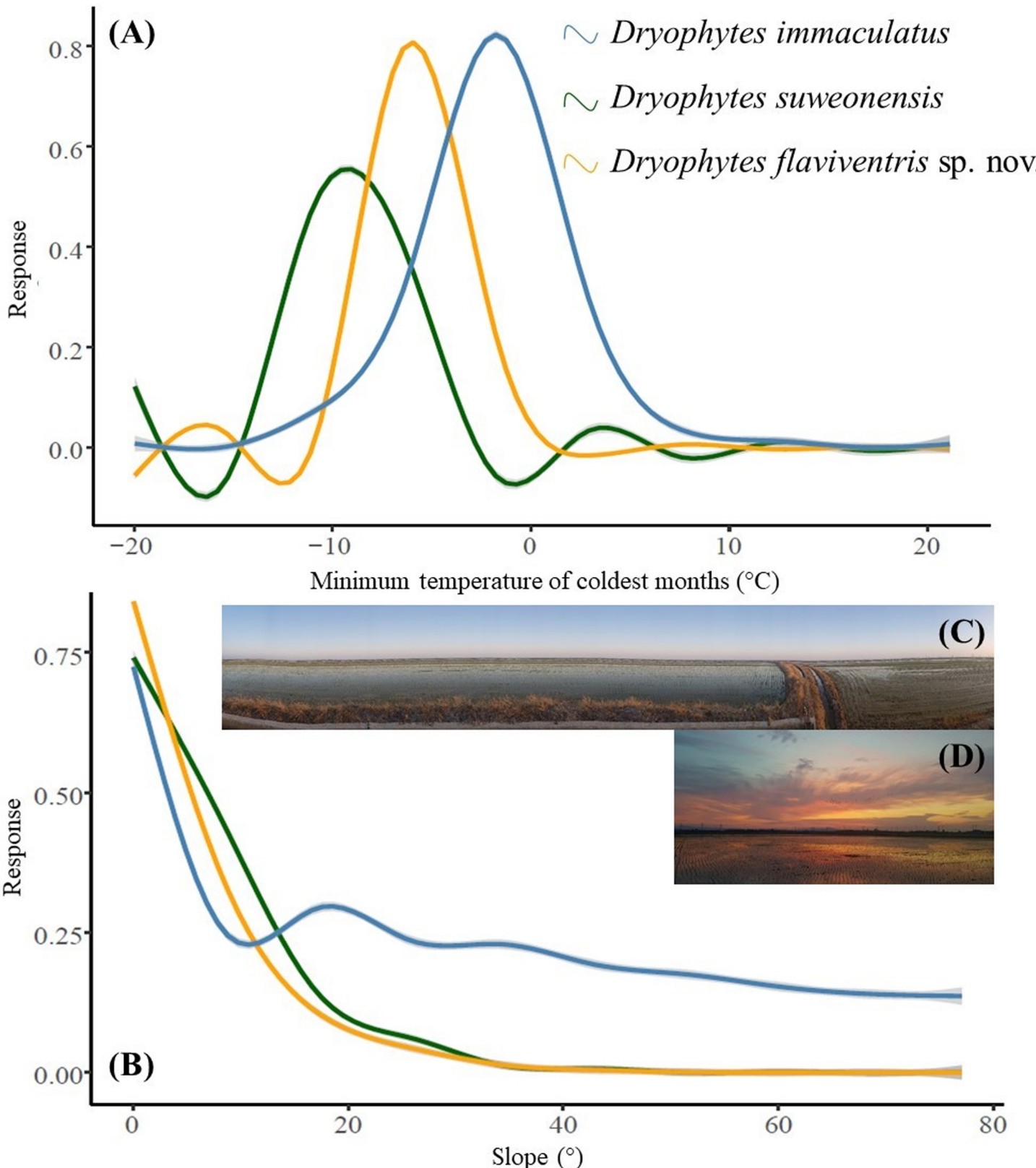

**Fig 9. Interpretation of jacknife analysis on habitat suitability to determine variables of importance for *Dryophytes immaculatus*, *D. suweonensis* and *D. flaviventris* sp. nov.** The minimum temperature of coldest month (Bio6; A) was found to be only variable significant for the three species in permutation tests, while slope (B) adequately represent landscape requirement for *D. flaviventris* sp. nov., illustrated in (C) and (D). Pictures from sites where the species was sampled in Iksan (C) and Buyeo (D).

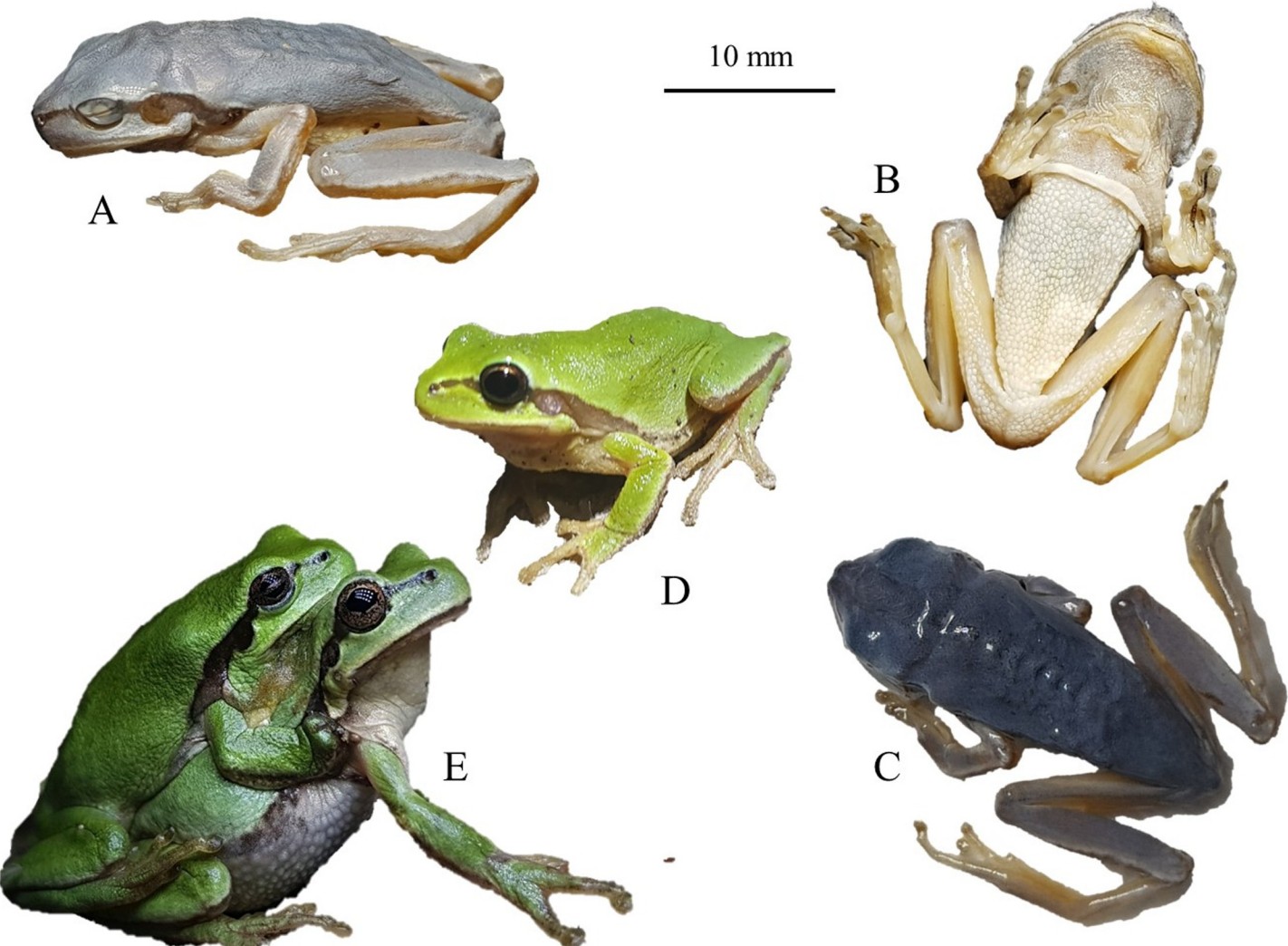

**Fig 10.** *Dryophytes flaviventris* sp. nov., holotype (A, B, C), different individuals in life (D) and in amplexus (E). The pictures of live individuals highlight the yellow coloration based on which the name was selected. The scale bar is for the holotype only.

**Nomenclatural history.** The nomenclatural history of the new species is connected to that of *D. suweonensis* (described in Suwon, R Korea), and therefore that of *D. immaculatus* (described in Shangai, P R China; [97]; holotype SMF 2310; [98]). *Hyla chinensis immaculata* [99, 100] was split from *H. chinensis* and then revised as *Hyla arborea immaculata* [101] a few years later, and therefore under the *H. arborea* group [102]. A comparison with individuals from the Asian mainland resulted in the epithet *H. immaculata*, first used by Stejneger [97]. *Dryophytes suweonensis* was described in 1980 [103] and the holotype is stored in Osaka Museum of Natural History (OMNH; Am 6035).

It is important to note that another species was described from the Korean Peninsula (*H. stepheni*, [104]), along with individuals from the Ussuri River [105] although later synonymised to *H. japonica* [106–108], after its formal description as *H. japonica* [109]. The possibility that *D. flaviventris* corresponds to *H. stepheni* is nullified by the original description stating that the "head is a little larger than that of *H. arborea* [= *D. japonicus*], broader than long", a

morphological trait to segregate *D. suweonensis* and *D. japonicus* [77, 103], with *D. suweonensis* more slender than the latter, and *D. flaviventris* being even more slender than *D. suweonensis*.

Additional species have been described further north than the range of any of the three species, and discussed here to confirm adequate taxonomy (see [103] for a longer version): *H. arborea japonica*, *H. a. immaculata*, *H. a. ussuriensis* and *H. stepheni* [110], and *H. sodei-campi* [111]. These were reduced to *H. a. immaculata* [112, 113], then to *H. a. japonica* [114], or *H. japonica* [115, 116]. The segregation between *D. suweonensis*, *D. immaculatus*, *H. ussuriensis and H. sodei-campi* is confirmed by Kuramoto [103] based on morphological cues extracted from Nikolskii [110] and Kostin [111], by Borzée [46] based on genetics and skeleton analyses, and finally by this study. The genus name *Hyla* was later amended to *Dryophytes* [56]. Therefore, to the best of our knowledge, there is no available scientific name for treefrogs in Jeolla Province (see also [117]).

**Etymology.**   We name this new species *Dryophytes flaviventris* sp. nov. The specific name "flaviventris" is a masculine noun used in apposition and based on the Latin words "flavus" (yellow) and "ventris", the genitive singular of venter (belly). The species name refers to the strong yellow marking on males, and the yellow hues on females (Fig 10). We suggest the English vernacular name "Yellow-bellied treefrog", the Korean common name 노랑배청개구리and the Chinese common name 黄腹雨蛙.

We recommend *D. flaviventris*, *D. suweonensis* and *D. immaculatus* to be collectively referred as the "*Dryophytes immaculatus* group" based on the seniority of the species description [100], in opposition to the *D. japonicus* group [52, 56]. To clarify the distinction with the other clades, we recommend the use of "Chinese immaculate treefrog" for *D. immaculatus* (无斑雨蛙 in Chinese and 민무늬청개구리 in Korean) as a way to distinguish with populations of *D. japonicus* occurring in the country [59], and for which taxonomy is yet unresolved [45]. The taxonomy of *D. suweonensis* is now likely to be stable, 수원청개구리 in Korean and 水原雨蛙 in Chinese.

**ZooBank registration.**   We hereby state that the present paper has been registered to the Official Register of Zoological Nomenclature (ZooBank) under LSID: urn:lsid:zoobank.org:pub:6155C475-42DA-4888-9F5A-A97EF67081DA. The new species name *Dryophytes flaviventris* sp. nov. has been registered under LSID: urn:lsid:zoobank.org:act:B26C2F1E-4E5F-432D-A95F-E661E95A7A02.

**Nomenclatural acts.**   The electronic edition of this article conforms to the requirements of the amended International Code of Zoological Nomenclature, and hence the new names contained herein are available under that Code from the electronic edition of this article. This published work and the nomenclatural acts it contains have been registered in ZooBank, the online registration system for the ICZN. The ZooBank LSIDs (Life Science Identifiers) can be resolved and the associated information viewed through any standard web browser by appending the LSID to the prefix "http://zoobank.org/". The LSID for this publication is: urn:lsid:zoobank.org:pub:6155C475-42DA-4888-9F5A-A97EF67081DA. The electronic edition of this work was published in a journal with an ISSN, and has been archived and is available from the following digital repositories: PubMed Central, LOCKSS.

## Discussion

Our results describe two important developments for the taxonomy of *Dryophytes* sp. in North East Asia. First, we resolve the question regarding the relationship between *D. immaculatus* and *D. suweonensis* [44–47] and demonstrate a clear distinction between the two species with a divergence estimated c. 1.02 mya. The two species belong to two segregated genetic clustering (Fig 1), with *D. immaculatus* present on the lowlands of eastern China, and *D. suweonensis* on

the lowlands of the western Korean Peninsula (Fig 1). Also, the call properties of the two species are different, and they can be identified by a higher number of independent pulses before the connected pulses in *D. suweonensis* than in *D. immaculatus* (Figs 1, 2 and 3). Furthermore, the two species display different head morphologies and limb length, with a clear difference in the presence of digital webbing in *D. suweonensis* and the almost absence in *D. immaculatus* (Fig 6). In term of morphology, the oral disk of tadpoles is also different, setting *D. immaculatus* aside from other North East Asian *Dryophytes* sp. for which data is available (Fig 7). Finally, the landscape requirements of the two species are also significantly different (Figs 8 and 9), with models showing a clear distinction and a latitudinal segregation, at the exception of an area around Beijing for *D. immaculatus*, matching with the distribution of the species [59, 61].

Second, we describe the presence of a new species from R Korea, *D. flaviventris*. The species is different from *D. suweonensis* based on genetics, estimated to have diverged c. 0.97 mya, with the two species pertaining to segregated genetic groups (Fig 2). The two species are geographically segregated by the Chilgap mountain range (Fig 1), and *D. flaviventris* is now known to occur in Buyeo, Nonsan and Iksan, while the extirpated populations in Wanju and Gunsan (R Korea) are also likely to have belonged to this species. In addition, the call properties of the two species are different, with the number of independent pulses also different between the two species (Figs 1–3). Finally, the morphological characteristics of *D. suweonensis* and *D. flaviventris* are not overlapping, even when corrected for SVL. In addition, the length of the webbing between the 2$^{nd}$ and 3$^{rd}$ toes is not overlapping in the samples collected (Fig 6). Finally, the two species also differ in landscape suitability, with the two species preferring flat and low elevation landscapes, but *D. suweonensis* being better suited for landscapes at higher latitude than *D. flaviventris*.

The results of the genetic analysis demonstrate the presence of *D. suweonensis* x *D. immaculatus* hybrids in the range of *D. immaculatus*, and the presence of *D. suweonensis* x *D. flaviventris* in the range of *D. flaviventris*. While genetic distance between species is not a clear criteria for species distinction [118], the pattern presented here is consistent with patterns of hybridisation in other *Dryophytes* species. For instance, *D. cinereus* and *D. gratiosus* hybridise at the edge of their ranges [119, 120], the same way *D. chrysoscelis* and *D. versicolor* do [121], and similarly to the sister genus in Europe (e.g. *Hyla arborea* and *H. intermedia*; [122]). Hybridisation may be unidirectional with all the *D. suweonensis* included in this study found to be pure, although a larger sample size would be needed to confirm this pattern. In addition, genetic data for the population in Beijing would provide information on the ability of either species to disperse north of the Yellow sea, an area of weak correspondence with habitat suitability for both of the northern species (Fig 8).

The relation between the three clades during the lowest point of the Yellow sea cannot be answered here, although it was more complex than today as the continental shelf was exposed [11] and the seashore was a few thousand kilometres further south than it is today [11, 62]. The lowest point in sea level was reached about 21 000 years BP [11], and the high sea level 7,000 years BP ([11]; see Fig 2 therein). Therefore, the last contact point between the three clades on the Yellow sea basin is a minimum of 23,000 years old and explains why evidences of hybridisation were found. Another barrier to the three clades during the LGM was the presence of wide paleorivers ([62]; Fig 1). Larger rivers have a strong barrier effect on the range of Hylid species, for instance the Vistula River segregates *H. orientalis* and *H. arborea* in Poland [7], the Dead Sea Rift segregates *H. felixarabica* and *H. savignyi* in Israel [123], the Garonne River segregates *H. arborea* and *H. meridionalis* in France [7], and the Mangyeong River is the southern limit of *D. flaviventris* [60]. Therefore, it is expected that the Paleo-Han, Huanghe and Yangtze Rivers (Fig 1) merged into a super-river, creating a barrier between *D. immaculatus* and the two other species. The contact zone between *D. suweonensis* and *D. flaviventris*

may have been further south along the continuity of the Chilgap mountain, potentially segregated by the paleo-Geum River (Fig 1). Despite not being one of the mountainous ranges with the highest elevation, the Chilgap mountain constitutes a non-crossable landscape element to these species, similarly to other landscape features on the Korean peninsula resulting in genetic isolation in *P. chosenicus* [124], *Hynobius* spp. [125, 126], and *D. japonicus* [45, 127]. Likewise, as the presence of *D. immaculatus* has not been confirmed south of the Yangtze river, it is likely that the species' southern limit was constricted to the northern drainage basin of the paleo-Yangtze River (Fig 1). Further, the barrier created by the paleo-Yangtze River was strengthened by the reinforcement of the East Asian Monsoon system since the mid-Pleistocene [128, 129], resulting in heavier rainfall [17, 130] and affecting the amount of water carried by rivers [35].

Alternatively, the three species may have been in contact further south in the area of the delta created by the paleo-river drainage system (Fig 1). Several progradational deltas were created during the sea level rise towards the Yellow sea basin [11], and were characterised by sediments brought by rivers [131], resulting in low angle slopes, such as preferred by the three species (Fig 9). While deltas may have been subjected to temporary brackish conditions, this should not have prevented the use of the habitat by the species due their resilience to relatively high saline concentration ([132]; and author's unpublished data: *D. suweonensis* breeding at salinity = 1042 ppm in Gangwha Island and *D. flaviventris* breeding in salinity = 987 ppm in Iksan).

In addition, the location of the current Yellow sea may have been the location of the refugia during the LGM. The frost line was further north, around current Beijing latitude, and the Korean Peninsula and Chinese eastern lowland were cold and dry, although not covered by glaciers [31, 32]. Other species were also present in refugia in the area, such as *Pelophylax nigromaculatus* [42] and *Onychodactylus koreanus* [133].

For marine animals in the Yellow sea, rising sea levels and range expansion resulted in genetic homogeneity and rapid population growth [134–137], while lowering of sea levels are associated with habitat fragmentation and potential genetic bottleneck [35, 138]. The $F_{ST}$ value determined here for *Dryophytes* sp. follows the same pattern, although inverted, as a result of the increase of sea level over the last 23,000 years. The three species were likely present on the Yellow sea basin but saw their range decrease from 21,000 y BP until 7,000 y BP, when ranges contracted even further in relation to the 8.2 ka cooling event [139], before stabilising. Similarly to our data, *D. suweonensis* has a lower genetic diversity than the sympatric *D. japonicus* [88], a species likely less affected by sea level variation due to its presence at higher altitude [140, 141]. In contrast, *D. suweonensis* and *D. flaviventris* (and likely *D. immaculatus*) are expected to have benefitted from rice agriculture over the last 1,000 years [95], potentially resulting in population expansion, followed by numerous local extirpation seen through missing haplotypes [142] as a result of landscape anthropisation [96].

The result of the landscape suitability analysis highlighted the minimum temperature of the coldest month (Bio6) as the only variable significant for all three species (Fig 9). While it may not seem to be an important variable as the species are hibernating in hibernaculum during the coldest month (e.g. [143]; [144]), it becomes important when related to the type of habitat used. The three species are occurring at low elevation in riparian flat wetlands only (e.g. median elevation of sites for *D. suweonensis* and *D. flaviventris* = 1 m a.s.l.), and they hibernate within the same habitat [94]. Thus, the hibernaculum available are not diverse, and it is difficult for the species to reach deep underground due to the absence of deep vertical structures (e.g. trees) along which they can bury themselves [145–147]. Therefore, slight variations in minimum temperature will result in a lowering of the frost line, such as it likely happened 8.2 kya [139], and in the need for specific cold tolerance mechanisms. While this has not been

studied in any of these species, the sister clade *D. japonicus* is known to tolerate colder temperatures in Russia (< -35˚C; [148]) than in Japan (> - 20˚C; [149]), highlighting variations in adaptive physiology to specific environments. Climate change will result in abrupt increases in temperature in some areas [150, 151], thus lowering the risk of freezing, although this is unlikely to be beneficial for the species as earlier emergence from hibernation before emergence of preys will result in starvation. In addition, Hylids generally breed directly after emergence [152], but as these three species rely on agricultural farming for breeding [91], and because farmers will plant rice later as it will grow faster under warmer climates [153] the decoupling of dynamics is likely to result in changes in the breeding output of these species, and likely in a lower recruitment. Finally, even if individuals manage to breed later, it will result in higher competition with species currently breeding a few weeks later, such as *Pelophylax nigromaculatus*, and the presence of more numerous predators following the arrival of migrating birds. In addition, climate change is likely to result in a northern shift of adequate habitats [154], a pattern that the species cannot follow.

Interestingly, while a difference in the skeleton of phalanges was detected between *D. immaculatus* and *D. flaviventris* [46, 92], no reason could be attributed to this variation. The results presented here show that the difference in skeleton is reflected by variations in webbing between these two species, and *D. suweonensis* (Fig 6). While it is here as well difficult to ascertain why there is a difference in morphology, it is likely related to the environment. Indeed, the three species breed in the same type of environment, but competition is different. *Dryophytes suweonensis* and *D. flaviventris* will principally compete with *D. japonicus* for microhabitat and calling spaces [93, 155–158], resulting in a special displacement in microhabitat use and the need for these two species to swim to reach the centre of wetlands [155]. On the other hand, *D. immaculatus* is not sympatric with any other Hylid species and will be principally competing for breeding space with *Fejervarya limnocharis* and *Microhyla fissipes* [59]. Because of the lower similarity between *D. flaviventris* and these two species in comparison to intra-Hylid competition, the pressure to use a different microhabitat is lower, and likely resulting in the absence of requirement for *D. immaculatus* to be a good swimmer. Interestingly though, males of the species were also found breeding while holding on vertical vegetation (Fig 1; see [155] for details on this behaviour). We therefore hypothesise here that there will be a difference in distance between calling perches and bank between these species, with *D. immaculatus* present closer to banks than the two other species.

The size variations between the three species highlighted by the PCA (Fig 5) may pertain to several pressures. First, the species may be drifting away from a common ancestor, independently of ecological pressures [159, 160]. More likely, the difference in pressure exerted on the three clades is the reason for divergence. As species that are most similar in body size compete the most strongly [161, 162], it is likely that the evolution in morphology for the two Korean species is driven by competition with *D. japonicus*. As the species with which *D. immaculatus* is competing are different, the evolutionary pressure will be different (see review by Dayan and Simberloff [163], and the availability of a wider variety of preys will result in variation in morphology, a trait clearly demonstrated in treefrogs [164–166] and shown in Caribbean tree frog (*Osteopilus* sp.; [167]). A similar pressure is expected to be the reason for the difference in tadpole morphology, although too little is known on tadpoles of this species to determine the reason for the loss of a tooth row in *D. immaculatus*.

Finally, as two of the five known populations for *D. flaviventris* are already extirpated, in Gunsan and Wanju (R Korea), it is important to conduct conservation assessments rapidly to prevent the extinction of this species occurring on a very narrow and declining range. The policy recommendation for the conservation of *D. suweonensis* also applies to *D. flaviventris* [168], and we urge local governments to take actions to designate protected areas for this

species, an urgent requirement for numerous species in R Korea [169]. This situation is however not restricted to *D. flaviventris* as only two populations of *D. immaculatus* could be found over 49 days of field work between 2017 and 2019, despite the species being assessed as abundant a few decades ago. A reason for this change in dynamics is likely the drying of the Yangtze River's valley, following the transformation of agricultural wetlands into dry agriculture because of the shift in food consumption from rice to wheat in China.

## Supporting information

**S1 Appendix. Dataset used for landscape modelling.** This dataset includes longitude and latitude data for each of the datapoints included in the data analysis presented.
(XLSX)

**S1 Fig.**
(JPG)

## Acknowledgments

We are grateful to Sungsik Kong, Daniel Macias, Timothy Bova, Kyongman Heo, Choog-ho Ham, and Yi Yang for their help during field work. We are also grateful to Mr Yu Sang Hong for his insistence in obtaining permits to study the populations in Iksan and Buyeo when we had not yet understood the significance of the populations in these areas. We are also thankful for the picture used in Fig 5. Finally, we thank Hollis Dahn for the picture used for Fig 5. We are also grateful to the Hanns Seidel Foundation for supporting research, communication and meetings in DPR Korea. The samples in Paju (R Korea) area were collected in 2013 under the Ministerial authorisation number 2013–16, while the other samples were collected in 2014 under the permits 2014–04, 2014–08 and 2014–20. Sampling in DPR Korea was conducted under the authorisation provided by the Ministry of Land and Environment Protection and sampling in PR China was conducted under the authorisation provided by Nanjing Forestry University. IACUC permits are not required when under ministerial authorisation for *Dryophytes suweonensis* and are not required for *Dryophytes immaculatus*.

## Author Contributions

**Conceptualization:** Amaël Borzée, Mi-Sook Min.

**Data curation:** Amaël Borzée, Kevin R. Messenger, Shinhyeok Chae, Desiree Andersen, Jordy Groffen, Ye Inn Kim, Junghwa An, Siti N. Othman, Kyongsin Ri, Tu Yong Nam, Yoonhyuk Bae, Jin-Long Ren, Jia-Tang Li, Ming-Feng Chuang, Yoonjung Yi, Yucheol Shin, Taejoon Kwon, Mi-Sook Min.

**Formal analysis:** Amaël Borzée, Kevin R. Messenger, Shinhyeok Chae, Desiree Andersen, Jordy Groffen, Ye Inn Kim, Junghwa An, Tu Yong Nam, Jia-Tang Li, Yikweon Jang, Mi-Sook Min.

**Funding acquisition:** Amaël Borzée, Yikweon Jang, Mi-Sook Min.

**Investigation:** Amaël Borzée, Kevin R. Messenger, Desiree Andersen, Kyongsin Ri, Yoonhyuk Bae, Jin-Long Ren, Ming-Feng Chuang, Yoonjung Yi, Yucheol Shin, Taejoon Kwon, Yikweon Jang, Mi-Sook Min.

**Methodology:** Amaël Borzée, Shinhyeok Chae, Desiree Andersen, Junghwa An, Siti N. Othman, Yoonhyuk Bae, Ming-Feng Chuang, Taejoon Kwon, Yikweon Jang, Mi-Sook Min.

**Project administration:** Amaël Borzée, Kevin R. Messenger, Kyongsin Ri, Mi-Sook Min.

**Resources:** Amaël Borzée, Kevin R. Messenger, Siti N. Othman, Mi-Sook Min.

**Software:** Amaël Borzée, Desiree Andersen, Siti N. Othman, Mi-Sook Min.

**Supervision:** Amaël Borzée, Kevin R. Messenger, Kyongsin Ri, Jia-Tang Li, Ming-Feng Chuang, Taejoon Kwon, Yikweon Jang, Mi-Sook Min.

**Validation:** Amaël Borzée, Kevin R. Messenger, Kyongsin Ri, Tu Yong Nam, Taejoon Kwon, Yikweon Jang, Mi-Sook Min.

**Visualization:** Amaël Borzée, Yikweon Jang, Mi-Sook Min.

**Writing – original draft:** Amaël Borzée, Desiree Andersen, Yikweon Jang, Mi-Sook Min.

**Writing – review & editing:** Amaël Borzée, Kevin R. Messenger, Shinhyeok Chae, Desiree Andersen, Jordy Groffen, Ye Inn Kim, Junghwa An, Siti N. Othman, Kyongsin Ri, Tu Yong Nam, Yoonhyuk Bae, Jin-Long Ren, Jia-Tang Li, Ming-Feng Chuang, Yoonjung Yi, Yucheol Shin, Taejoon Kwon, Mi-Sook Min.

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
