## [Decision Letter · Decision Letter 0]

29 Jan 2020

PONE-D-19-29311

Yellow sea mediated segregation between North East Asian Dryophytes species

PLOS ONE

Dear Pr. Borzée,

Thank you for submitting your manuscript to PLOS ONE. After careful consideration, we feel that it has merit but does not fully meet PLOS ONE’s publication criteria as it currently stands. Therefore, we invite you to submit a revised version of the manuscript that addresses the points raised during the review process.

We would appreciate receiving your revised manuscript by Mar 14 2020 11:59PM. To enhance the reproducibility of your results, we recommend that if applicable you deposit your laboratory protocols in protocols.io, where a protocol can be assigned its own identifier (DOI) such that it can be cited independently in the future. For instructions see: http://journals.plos.org/plosone/s/submission-guidelines#loc-laboratory-protocols

We look forward to receiving your revised manuscript.

Kind regards,

Si-Min Lin, Ph.D.

Academic Editor

PLOS ONE

Journal Requirements:

3. In your Methods section, please provide additional location information of the collection sites, including geographic coordinates for the data set if available.

4. Thank you for including your ethics statement: "The samples in Paju (R Korea) area were collected in 2013 under the Ministerial authorisation number 2013-16, while the other samples were collected in 2014 under the permits 2014-04, 2014-08 and 2014-20. Sampling in DPR Korea was conducted under the authorisation provided by the Ministry of Land and Environment Protection and sampling in PR China was conducted under the authorisation provided by Nanjing Forestry University. IACUC permits are not required when under ministerial authorisation for Dryophytes suweonensis and are not required for Dryophytes immaculatus."

a. Please amend your current ethics statement to include the full name of the ethics committee that approved your specific study (re. animal capture)

For additional information about PLOS ONE submissions requirements for ethics oversight of animal work, please refer to http://journals.plos.org/plosone/s/submission-guidelines#loc-animal-research

5. We note that you are reporting an analysis of a microarray, next-generation sequencing, or deep sequencing data set. PLOS requires that authors comply with field-specific standards for preparation, recording, and deposition of data in repositories appropriate to their field. Please upload these data to a stable, public repository (such as ArrayExpress, Gene Expression Omnibus (GEO), DNA Data Bank of Japan (DDBJ), NCBI GenBank, NCBI Sequence Read Archive, or EMBL Nucleotide Sequence Database (ENA)). In your revised cover letter, please provide the relevant accession numbers that may be used to access these data. For a full list of recommended repositories, see http://journals.plos.org/plosone/s/data-availability#loc-omics or http://journals.plos.org/plosone/s/data-availability#loc-sequencing.

This work was supported by a Conservation Research grant from The Biodiversity Foundation to AB, a research grant from the National Research Foundation of Korea (2017R1A2B2003579) to YJ, and by a grant from the National Institute of Biological Resources (NIBR), funded by the Ministry of Environment (MOE) of the Republic of Korea (NIBR201803101) to MSM.

7. We note you have included a table to which you do not refer in the text of your manuscript. Please ensure that you refer to Table 3 in your text; if accepted, production will need this reference to link the reader to the Table.

Additional Editor Comments:

Dear Dr. Borzee and Dr. Min,

Sorry for the long wait and thanks for your patience. One of our reviewers delayed his task for quite long period so that we had to find a new one to replace his task; making the review process delayed for quite a long time.

First I would like to congratulation for your excellent work to clarify the complicated interrelationship among these morphologically similar taxa. I would expect this work to be published and become one of the classic research both in this region and in this taxon.

However, both reviewers have proposed some weakness about the current manuscript. I agree most of their comments and think the manuscript needs some editing. Major problems were raised concerning the details about RAD-seq protocol, and the expression of the figures. The major comments include:

1. The first reviewers has proposed some comments about adding the phylogenetic tree as an independent figure; and also provided some consideration about RAD-seq analyses.

2. Too much information was intended to be included in Fig. 1, 2, and 3. Both reviewers and me noticed this problem and felt these three figures too “crowded”. Since there is no page limitation for PLOS ONE, I think you could just feel free to increase the number of figures and decrease the crowdedness in these figures.

2. Specific comments to Fig. 1: (1) The phylogeny of the three species could be independent to a new figure, as suggested by the first reviewer. (2) Photographs of the frogs could be moved to some other figures. The three photos used here did not provide taxonomic diagnosticity; therefore, they are not effectively informative for unfamiliar readers. (3) A black outline appears close to Anhui Province, is that necessary? What does that mean? (4) Coloration usage of the three species, except for D. immaculatus, is obscure. The three greenish and yellowish colors are hard to be distinguished by color-blind people, and are also difficult for normal readers.

3. Specific comments to Fig. 2: The photographs and the sonograms should not overlap with the Structure assignment. For example, the photographs could be moved upward, and the sonograms could be moved downward.

4. Specific comment to Fig. 3: The sonograms could be moved out from the PCA scattered plot to reduce overlapping. Finally, please consider the comments from the reviewers that the sonograms have appeared repetitively too many times.

Most of the comments above are editing and writing problems. I wish these comments could help to make this manuscript a more “reader-friendly” paper without altering its original scientific values. I look forward to see the revised version, which must be worthy to be published in the very near future.

With best wishes,

Si-Min LIN

Reviewers' comments:

Reviewer's Responses to Questions

**Comments to the Author**

1. Is the manuscript technically sound, and do the data support the conclusions?

Reviewer #1: Yes

Reviewer #2: Partly

2. Has the statistical analysis been performed appropriately and rigorously? 

Reviewer #1: N/A

Reviewer #2: Yes

3. Have the authors made all data underlying the findings in their manuscript fully available?

Reviewer #1: No

Reviewer #2: No

4. Is the manuscript presented in an intelligible fashion and written in standard English?

Reviewer #1: Yes

Reviewer #2: No

5. Review Comments to the Author

Reviewer #1: The study of Borzée et al. focused to solve Dryophytes species complex in North East Asia region by utilizing multiple evidences: genetic, morphology, communicate signal and niche modeling. They also tried to infer impact of past fluctuant of sea level on the evolution of this group. The data did support the taxonomic status of this group and new species proposed by the authors. However, there is still a number of major concerns. Please find the detailed comments below.

1. Sample info: Number of samples among analyses (RAD-seq, morphology, call properties) were not consistent which causes confusing. It would be better if you have a table to sum up you the information (samples, location, species…) of the dataset for each analysis.

2. Figures: I found some figures were hard to read, especially the figure 1 where you tried to push too much information into it. Further, some details were repeated such as the call frequencies were in three different figures (figs 1-3), while other important information was ignored. For me, the phylogeny tree from RAD-seq data which should be showed in full detail in the manuscript or at least in supplementary.

3. Niche modeling: Using all the variables from Worldclim could lead to bias due to high correlated among variables, I suggest to check other studies that tested niche modeling on similar taxa (frogs) to select suitable bioclimatic variables or use some tool such as ENMTools to exclude the ones that are high correlated.

4. RAD-seq: Despite that RAD-seq is the most popular method to generate massive data for non-model species, I think in this study this part is the weakest part with the least effort or explaining detail. This method has caveats, like allele dropout that can lead to misestimate genetic divergence and diversity (Gautier et al. 2013; Schweyen, Rozenberg, and Leese 2014). This bias was totally ignored in the manuscript. Moreover, although authors used one of the best options for RAD-seq inference (STACKS pipeline), without reference genome inferring the different sets of pipeline’s parameter in order to find the best set for the targeted species is highly recommended (Catchen et al. 2013; Harvey et al. 2015; Mastretta-Yanes et al. 2015; Paris, Stevens, and Catchen 2017). Finally, I think utilizing the massive data from RAD-seq to only infer the phylogenetic relationship and genetic structure is quite wasted. This dataset is capable to infer more detail of historical demography of targeted species.

5. Data accessibility: Even though the authors stated that the relevant data are available, the data of RAD-seq and others are nowhere to be found.

Reviewer #2: Overall, I think this is an interesting and comprehensive study. The authors studied the patterns of genetic variation, acoustic features, morphology and external morphological differentiation among populations of Dryophytes species around the Yellow sea. My general impression is that the methods could be presented more clearly and should discussed more carefully. Although, the study is well-suitable for publication in Plos One, I have some critics that the authors should consider for a revision.

Major comments:

I think the introduction and discussion parts may need some re-working. The aim of this study is not clear—I am not completely sure whether this is simply a study of species delimitation or a phylogeographic study. The authors used a long paragraph talking about the potential effects of sea level fluctuations on population differentiation and distributions, however, they didn’t estimate divergence times between species or historical changes of population size. I think it’s hard to evaluate the potential effect of sea level fluctuations on species differentiation without estimating these parameters.

- All the analyses in this study were carried out based on three pre-defined ‘clades’, but the definition of these three clades is missing. This information should provide in the beginning of the material and methods part.

-L508-520: I would recommend the authors to add genetic analysis designed for hybridization detection (e.g. NewHybrids).

It would be interesting if the authors can provide some morphological and acoustic information of these hybrid individuals.

Other comments:

- Are individuals used in genetic analysis also been used in acoustics/morphology analyses? It is hard to understand how many samples from which populations have been investigated. A summary table might help clarify.

-L312-313: what’s the meaning of the numbers 303, 533 and 333?

I don’t feel qualiﬁed to judge the English, as it is not my mother tongue; however, I do feel that in some parts the English is not up to standard and is sometimes rather ambiguous. I think a thorough revision by a native English proofreader would increase the readability of this article.

---

## [Author Response · Author response to Decision Letter 0]

17 Apr 2020

Journal Requirements:

We have reformatted the manuscript according to the journal’s guidelines. Please find the updated version uploaded on the website.

We have modified the name of the file in the text and added the caption, such as: 

Appendix 1 (S1 Appendix): Dataset used for landscape modelling. This dataset include longitude and latitude data for each of the datapoints included in the data analysis presented.

3. In your Methods section, please provide additional location information of the collection sites, including geographic coordinates for the data set if available.

We have added GPS coordinates to each sampling locality, although maintaining a low accuracy to ensure the protection of the species.

4. Thank you for including your ethics statement: "The samples in Paju (R Korea) area were collected in 2013 under the Ministerial authorisation number 2013-16, while the other samples were collected in 2014 under the permits 2014-04, 2014-08 and 2014-20. Sampling in DPR Korea was conducted under the authorisation provided by the Ministry of Land and Environment Protection and sampling in PR China was conducted under the authorisation provided by Nanjing Forestry University. IACUC permits are not required when under ministerial authorisation for Dryophytes suweonensis and are not required for Dryophytes immaculatus."

a. Please amend your current ethics statement to include the full name of the ethics committee that approved your specific study (re. animal capture)

No IACUC permits were required.

For additional information about PLOS ONE submissions requirements for ethics oversight of animal work, please refer to http://journals.plos.org/plosone/s/submission-guidelines#loc-animal-research

No IACUC permits were required.

5. We note that you are reporting an analysis of a microarray, next-generation sequencing, or deep sequencing data set. PLOS requires that authors comply with field-specific standards for preparation, recording, and deposition of data in repositories appropriate to their field. Please upload these data to a stable, public repository (such as ArrayExpress, Gene Expression Omnibus (GEO), DNA Data Bank of Japan (DDBJ), NCBI GenBank, NCBI Sequence Read Archive, or EMBL Nucleotide Sequence Database (ENA)). In your revised cover letter, please provide the relevant accession numbers that may be used to access these data. For a full list of recommended repositories, see http://journals.plos.org/plosone/s/data-availability#loc-omics or http://journals.plos.org/plosone/s/data-availability#loc-sequencing.

The data was submitted to the European Nucleotide Archive (accession number PRJEB36680). Please see the detailed answer below for details.

This work was supported by a Conservation Research grant from The Biodiversity Foundation to AB, a research grant from the National Research Foundation of Korea (2017R1A2B2003579) to YJ, and by a grant from the National Institute of Biological Resources (NIBR), funded by the Ministry of Environment (MOE) of the Republic of Korea (NIBR201803101) to MSM.

We have updated the sentence suggested, such as:

This work was supported by a Conservation Research grant from The Biodiversity Foundation to AB, a research grant from the National Research Foundation of Korea (2017R1A2B2003579) to YJ, and by a grant from the National Institute of Biological Resources (NIBR), funded by the Ministry of Environment (MOE) of the Republic of Korea (NIBR201803101) to MSM and TK.

We have not deleted the sentence from the ms as the online form does not allow such an elaborate wording with multiple recipients for a grant.

7. We note you have included a table to which you do not refer in the text of your manuscript. Please ensure that you refer to Table 3 in your text; if accepted, production will need this reference to link the reader to the Table.

There may be a mistake here, Table 3 is inserted in the text, L. 249-250: “To be able to analyse morphometric variations without any bias due to the size of the individuals, we adjusted the dataset by dividing each value by the SVL of the individual. Variables in the dataset were generally correlated (Pearson’s correlation; n = 32; Table 3) so we used a PCA here as well to analyse variations between each of the clades”.

Additional Editor Comments:

Dear Dr. Borzee and Dr. Min,

Sorry for the long wait and thanks for your patience. One of our reviewers delayed his task for quite long period so that we had to find a new one to replace his task; making the review process delayed for quite a long time.

First I would like to congratulation for your excellent work to clarify the complicated interrelationship among these morphologically similar taxa. I would expect this work to be published and become one of the classic research both in this region and in this taxon.

Thank you for the editorial work, and we would like to apologise for the delay in resubmission as well. We have added a large number of individuals to the morphometric and genetic datasets, and running additional analyses took some time. 

However, both reviewers have proposed some weakness about the current manuscript. I agree most of their comments and think the manuscript needs some editing. Major problems were raised concerning the details about RAD-seq protocol, and the expression of the figures. The major comments include:

1. The first reviewers has proposed some comments about adding the phylogenetic tree as an independent figure; and also provided some consideration about RAD-seq analyses.

We have added a phylogenetic tree with divergence dating, and improved the text on the analyses. Please see the detailed answer to Reviewer #1’s comments.

2. Too much information was intended to be included in Fig. 1, 2, and 3. Both reviewers and me noticed this problem and felt these three figures too “crowded”. Since there is no page limitation for PLOS ONE, I think you could just feel free to increase the number of figures and decrease the crowdedness in these figures.

2. Specific comments to Fig. 1: (1) The phylogeny of the three species could be independent to a new figure, as suggested by the first reviewer. (2) Photographs of the frogs could be moved to some other figures. The three photos used here did not provide taxonomic diagnosticity; therefore, they are not effectively informative for unfamiliar readers.

We have created a new figure with the phylogenetic tree, and removed “crowding” from the other figures by moving the figures. Please find the new manuscript attached.

(3) A black outline appears close to Anhui Province, is that necessary? What does that mean? 

We mention the province in the manuscript and argue that it is important to indicate where the samples come from. 

(4) Coloration usage of the three species, except for D. immaculatus, is obscure. The three greenish and yellowish colors are hard to be distinguished by color-blind people, and are also difficult for normal readers.

We have adjusted the contrast of the colours used, it is clear now, even when printed in black and white. Please see the new figures attached.

3. Specific comments to Fig. 2: The photographs and the sonograms should not overlap with the Structure assignment. For example, the photographs could be moved upward, and the sonograms could be moved downward.

We have moved the sonograms and photographs so that they don’t overlap with the edge of any structure bar that would be not be assigned to a single clade as recommended. Please see the new figure.

4. Specific comment to Fig. 3: The sonograms could be moved out from the PCA scattered plot to reduce overlapping. Finally, please consider the comments from the reviewers that the sonograms have appeared repetitively too many times.

Here we argue that the sonograms do not overlap with any of the value on the graph and are informative as they link variables on a graph and something relevant to the behaviour of the species. We agree they are on three figures, on purpose, as they convey some visual continuity for the readers to understand more easily. It’s a long ms, and we would like to preserve reader’s interest all along the text. We agree that a black and white figure is not clear here and have added a note in the caption in this regard: “The colour version is needed to match clouds of points and call properties”.

Most of the comments above are editing and writing problems. I wish these comments could help to make this manuscript a more “reader-friendly” paper without altering its original scientific values. I look forward to see the revised version, which must be worthy to be published in the very near future.

Thank you, please find detailed answers to the points raised by the reviewers below as well as the revised versions attached to this letter.

With best wishes,

Si-Min LIN

Reviewers' comments:

Reviewer's Responses to Questions

Comments to the Author

Reviewer #1: 

The study of Borzée et al. focused to solve Dryophytes species complex in North East Asia region by utilizing multiple evidences: genetic, morphology, communicate signal and niche modeling. They also tried to infer impact of past fluctuant of sea level on the evolution of this group. The data did support the taxonomic status of this group and new species proposed by the authors. However, there is still a number of major concerns. Please find the detailed comments below.

Thank you for your time reviewing our manuscript, we appreciate the value of the comments and it made our work better. Please find the detailed answer to your comments below.

1. Sample info: Number of samples among analyses (RAD-seq, morphology, call properties) were not consistent which causes confusing. It would be better if you have a table to sum up you the information (samples, location, species…) of the dataset for each analysis.

We have added a summary table including this information, along with the sample voucher names for traceability. Is it table 1, such as:

Table 1: Sampling summary table. The sample sizes for each clade used for the genetic analyses, call properties, and morphometrics are summarised here (A). The individuals for which DNA was extracted and RAD-seq data submitted to the European Nucleotide Archive (accession number PRJEB36680) are also listed in this table (B).

(A) Samples size summary 

Species Genetics Calls Morphometrics

Dryophytes immaculatus 4 12 8

Dryophytes suweonensis 8 28 33

Dryophytes flaviventris sp. nov. 6 16 14

(B) Origin samples for RAD-seq data (European Nucleotide Archive accession number PRJEB36680)

Species Country of origin Locality Voucher ID (alias)

Dryophytes suweonensis Republic of Korea Pyeongtaek mms6883_HYLSU

Dryophytes flaviventris sp. nov. Republic of Korea Iksan mms8551_HYLFL

Dryophytes flaviventris sp. nov. Republic of Korea Iksan mms8552_HYLFL

Dryophytes flaviventris sp. nov. Republic of Korea Iksan mms8553_HYLFL

Dryophytes immaculatus People's Republic of China Anhui mms8665_HYLIM

Dryophytes immaculatus People's Republic of China Anhui mms8670_HYLIM

Dryophytes suweonensis Republic of Korea Pyeongtaek mms6884_HYLSU

Dryophytes suweonensis Republic of Korea Pyeongtaek mms6885_HYLSU

Dryophytes immaculatus People's Republic of China Anhui mms8666_HYLIM

Dryophytes immaculatus People's Republic of China Anhui mms8667_HYLIM

Dryophytes suweonensis Republic of Korea Eumseong mms4972_HYLSU

Dryophytes suweonensis Republic of Korea Eumseong mms4973_HYLSU

Dryophytes suweonensis Republic of Korea Eumseong mms4974_HYLSU

Dryophytes suweonensis Republic of Korea Eumseong mms5027_HYLSU

Dryophytes suweonensis Republic of Korea Eumseong mms5029_HYLSU

Dryophytes flaviventris sp. nov. Republic of Korea Iksan mms8548_HYLFL

Dryophytes flaviventris sp. nov. Republic of Korea Iksan mms8549_HYLFL

Dryophytes flaviventris sp. nov. Republic of Korea Iksan mms8550_HYLFL

2. Figures: I found some figures were hard to read, especially the figure 1 where you tried to push too much information into it. Further, some details were repeated such as the call frequencies were in three different figures (figs 1-3), while other important information was ignored. For me, the phylogeny tree from RAD-seq data which should be showed in full detail in the manuscript or at least in supplementary.

We have removed the phylogenetic tree and pictures from Fig 1, and created a new figure (Fig. 3) with this information. We decided to keep the call frequency on Fig 1, 2 and 4 as they convey some visual continuity for the readers to understand more easily. It’s a long ms, and we would like to preserve reader’s interest all along the text.

3. Niche modeling: Using all the variables from Worldclim could lead to bias due to high correlated among variables, I suggest to check other studies that tested niche modeling on similar taxa (frogs) to select suitable bioclimatic variables or use some tool such as ENMTools to exclude the ones that are high correlated.

We agree with the reviewer that testing for correlation is generally required, however, we did not know enough about the species to do so here, and decreasing the number of variables may have 1) potentially excluded variables of interest and 2) prevented us from comparing the importance of variables in case of mismatch of best-fitting models. In addition, no baseline models testing the effects of all bioclimatic variables on the species studied here has been published and our analyses can also be used for such a purpose for future models. We have added theses justifications to the manuscript such as (Lines 286-297): “While some variables selected may have been correlated, the ecology of D. immaculatus is relatively unknown and some reports hint at a potential presence at higher altitudes (http://www.amphibiachina.org/species/307). In addition, baseline ecological niche modelling has not yet been conducted for these species and variables of interest are still questionable. Therefore, to avoid the exclusion of relevant variables (Tytar et al., 2018), but also to create a baseline for future studies, we decided to include all bioclimatic variables and thus ensure the absence of preconceived bias on the ecological variables relevant to a species not yet described. In this framework, the inclusion of all variables further allowed us to compare the importance of each variables amongst species while a selection would have resulted in the use of different best-fit models for each species and it would have therefore prevented us from comparing the response variables”. 

4. RAD-seq: Despite that RAD-seq is the most popular method to generate massive data for non-model species, I think in this study this part is the weakest part with the least effort or explaining detail. This method has caveats, like allele dropout that can lead to misestimate genetic divergence and diversity (Gautier et al. 2013; Schweyen, Rozenberg, and Leese 2014). This bias was totally ignored in the manuscript. Moreover, although authors used one of the best options for RAD-seq inference (STACKS pipeline), without reference genome inferring the different sets of pipeline’s parameter in order to find the best set for the targeted species is highly recommended (Catchen et al. 2013; Harvey et al. 2015; Mastretta-Yanes et al. 2015; Paris, Stevens, and Catchen 2017). Finally, I think utilizing the massive data from RAD-seq to only infer the phylogenetic relationship and genetic structure is quite wasted. This dataset is capable to infer more detail of historical demography of targeted species.

Following the recommendation of the reviewer, we re-evaluated the parameters used for the ddRAD-seq data analysis. We assessed both metrics and variables based on the reference suggested (Paris, et al., 2017). When investigating the four major parameters in STACKS (r, m, M, n), we refined our parameters to a more stringent criteria as shown below, resulting in about 1/3 of refined SNPs (from 14,189 down to 5,819) for the analysis. We have edited the text of the manuscript accordingly: “. We further optimised the parameters of ‘ustacks’ to m = 4, M = 3 and n = 4, following the recommendation of the r80 method by Paris et al. (2017). Next, we selected specific population reads for which more than 50 % of loci were variable (-r 0.5)”.

However, the STRUCTURE analysis showed minimal changes in our revised result compared to the previous result. So we think that our RAD-seq data is stable enough to describe the genetic structure of three Dryophytes species we described in this manuscript.

We agreed that we used RAD-seq data in a limited manner, and it would be great to utilize to infer another genetic perspective of Dryophytes and Hyla species in East Asia, such as the historical demography. However, we believed that the number of samples used in the current study, and especially outside the Republic of Korea, is too low to utilize the full power of RAD-seq in such analysis.

5. Data accessibility: Even though the authors stated that the relevant data are available, the data of RAD-seq and others are nowhere to be found.

We submitted all our RAD-seq raw data to the European Nucleotide Archive (ENA) with the accession number PRJEB36680 (https://www.ebi.ac.uk/ena/data/view/PRJEB36680)

Reviewer #2: 

Overall, I think this is an interesting and comprehensive study. The authors studied the patterns of genetic variation, acoustic features, morphology and external morphological differentiation among populations of Dryophytes species around the Yellow sea. My general impression is that the methods could be presented more clearly and should discussed more carefully. Although, the study is well-suitable for publication in Plos One, I have some critics that the authors should consider for a revision.

Major comments:

I think the introduction and discussion parts may need some re-working. The aim of this study is not clear—I am not completely sure whether this is simply a study of species delimitation or a phylogeographic study. The authors used a long paragraph talking about the potential effects of sea level fluctuations on population differentiation and distributions, however, they didn’t estimate divergence times between species or historical changes of population size. I think it’s hard to evaluate the potential effect of sea level fluctuations on species differentiation without estimating these parameters.

We have re-formulated the last sentence of the discussion to clarify that the section on the impact of the Yellow Sea only is a discussion. It is now written such as: “Finally, we connect the impact of the Yellow sea level variations on the relationship of these three species”. The reason for not conducting additional analyses is that it may be inaccurate to go further than the new analyses on estimated divergence time with the current RAD-seq data without a proper outgroup – please see the new version of the ms with divergence estimate on Fig. 3 and updated STRUCTURE results on Fig. 2. We included D. japonicus RAD-seq data but the number of loci was significantly reduced because of the genetic divergence between the two groups, and going further would weaken the power of the analysis conducted here.

- All the analyses in this study were carried out based on three pre-defined ‘clades’, but the definition of these three clades is missing. This information should provide in the beginning of the material and methods part.

We have defined “clade” in the first sentence of the materials and methods such as recommended. It is now written such as: “Data for the three clades, defined as potentially divergent groups of individuals with different common ancestors, was collected between 2016 and 2019 in the Republic of Korea”.

-L508-520: I would recommend the authors to add genetic analysis designed for hybridization detection (e.g. NewHybrids). It would be interesting if the authors can provide some morphological and acoustic information of these hybrid individuals.

While this may be an interesting point to raise, we argue that it would not be falling within the framework of this manuscript and we prefer not addressing the subject of hybridization here.

Other comments:

- Are individuals used in genetic analysis also been used in acoustics/morphology analyses? It is hard to understand how many samples from which populations have been investigated. A summary table might help clarify.

The individuals are not the same, and we have added a table to clarify this points. Is it table 1, such as:

Table 1: Sampling summary table. The sample sizes for each clade used for the genetic analyses, call properties, and morphometrics are summarised here (A). The individuals for which DNA was extracted and RAD-seq data submitted to the European Nucleotide Archive (accession number PRJEB36680) are also listed in this table (B).

(A) Samples size summary 

Species Genetics Calls Morphometrics

Dryophytes immaculatus 4 12 8

Dryophytes suweonensis 8 28 33

Dryophytes flaviventris sp. nov. 6 16 14

(B) Origin samples for RAD-seq data (European Nucleotide Archive accession number PRJEB36680)

Species Country of origin Locality Voucher ID (alias)

Dryophytes suweonensis Republic of Korea Pyeongtaek mms6883_HYLSU

Dryophytes flaviventris sp. nov. Republic of Korea Iksan mms8551_HYLFL

Dryophytes flaviventris sp. nov. Republic of Korea Iksan mms8552_HYLFL

Dryophytes flaviventris sp. nov. Republic of Korea Iksan mms8553_HYLFL

Dryophytes immaculatus People's Republic of China Anhui mms8665_HYLIM

Dryophytes immaculatus People's Republic of China Anhui mms8670_HYLIM

Dryophytes suweonensis Republic of Korea Pyeongtaek mms6884_HYLSU

Dryophytes suweonensis Republic of Korea Pyeongtaek mms6885_HYLSU

Dryophytes immaculatus People's Republic of China Anhui mms8666_HYLIM

Dryophytes immaculatus People's Republic of China Anhui mms8667_HYLIM

Dryophytes suweonensis Republic of Korea Eumseong mms4972_HYLSU

Dryophytes suweonensis Republic of Korea Eumseong mms4973_HYLSU

Dryophytes suweonensis Republic of Korea Eumseong mms4974_HYLSU

Dryophytes suweonensis Republic of Korea Eumseong mms5027_HYLSU

Dryophytes suweonensis Republic of Korea Eumseong mms5029_HYLSU

Dryophytes flaviventris sp. nov. Republic of Korea Iksan mms8548_HYLFL

Dryophytes flaviventris sp. nov. Republic of Korea Iksan mms8549_HYLFL

Dryophytes flaviventris sp. nov. Republic of Korea Iksan mms8550_HYLFL

-L312-313: what’s the meaning of the numbers 303, 533 and 333?

Thank you for pointing out these typos. They are the number of notes analysed for each species, defined at the beginning of the section, and now correctly reported such as 302, 530 ad 333.

I don’t feel qualiﬁed to judge the English, as it is not my mother tongue; however, I do feel that in some parts the English is not up to standard and is sometimes rather ambiguous. I think a thorough revision by a native English proofreader would increase the readability of this article.

Some of the native English speaking co-authors have checked the ms. Thank you for pointing it out.

---

## [Editor Report · Decision Letter 1]

28 Apr 2020

PONE-D-19-29311R1

Yellow sea mediated segregation between North East Asian Dryophytes species

PLOS ONE

Dear Dr. Borzee and Dr. Min,

Thank you for submitting your manuscript to PLOS ONE. After careful consideration, we feel that it has merit but does not fully meet PLOS ONE’s publication criteria as it currently stands. Therefore, we invite you to submit a revised version of the manuscript that addresses the points raised during the review process.

We would appreciate receiving your revised manuscript by Jun 12 2020 11:59PM. To enhance the reproducibility of your results, we recommend that if applicable you deposit your laboratory protocols in protocols.io, where a protocol can be assigned its own identifier (DOI) such that it can be cited independently in the future. For instructions see: http://journals.plos.org/plosone/s/submission-guidelines#loc-laboratory-protocols

We look forward to receiving your revised manuscript.

Kind regards,

Si-Min Lin, Ph.D.

Academic Editor

PLOS ONE

Additional Editor Comments (if provided):

Dear Dr. Borzée and Dr. Min,

I am glad to read the revised version and found that most comments from the previous reviewers have been revised. In order to facilitate the timing of publication, your manuscript does not need another round of review by the original referees. Nevertheless, there are a few minor problems which need to be revised before the manuscript could be accepted. I would be more than happy to see the paper to be published as soon as possible if these editing problems could be solved.

Fig. 1. In the previous comments, we have proposed the redundancy of the outline of Anhui Province. After consideration, I still felt it redundant and even misleading. Hefei and Chuzhou appear at the upper part of Anhui Province, whereas the label of “Anhui” appears at the lower part, and this makes the readers confused. In this case, the precise position of the province is not important information for the readers. I suggest to eliminate the outline, and label the two localities as “Hefei, Anhui Prov.” and “Chuzhou, Anhui Prov.” This solves most the problems.

Fig. 1. As proposed by our previous comment, some colors used in the figure were similar. The major problem raised from colors of D. suweonensis and D. flaviventris; these two are not easily distinguished by color-blinded people. Therefore, please change one of them.

Fig. 2. I think you submitted the old version; please update. The photos, the sonograms, and the labels are still overlapping with the STRUCTURE probabilities.

Materials and methods, lines 103 – 104:

I suspected that you have misunderstood the meaning of the reviewer. There was a missing link about how you identify, and how you assign your samples to these three clades. Therefore, the reviewer was asking about the definition of “the three clades”, not the definition of “what a clade means”. I think you need a few sentences to clarify how you distinguish, how you identify, and how you assign the samples to the three clades. By sample locations? Morphology? Or some other pre-tests?

Minor revisions

Line 89: lack of a full stop at the end of the sentence.

Line 1242: “includes”

Sincerely yours,

Si-Min LIN, 2020/4/27

---

## [Author Response · Author response to Decision Letter 1]

5 May 2020

Dear Dr. Borzée and Dr. Min,

I am glad to read the revised version and found that most comments from the previous reviewers have been revised. In order to facilitate the timing of publication, your manuscript does not need another round of review by the original referees. Nevertheless, there are a few minor problems which need to be revised before the manuscript could be accepted. I would be more than happy to see the paper to be published as soon as possible if these editing problems could be solved.

=> Dear Editor, thank you very much for the support, we have now resubmitted the corrected version online.

Fig. 1. In the previous comments, we have proposed the redundancy of the outline of Anhui Province. After consideration, I still felt it redundant and even misleading. Hefei and Chuzhou appear at the upper part of Anhui Province, whereas the label of “Anhui” appears at the lower part, and this makes the readers confused. In this case, the precise position of the province is not important information for the readers. I suggest to eliminate the outline, and label the two localities as “Hefei, Anhui Prov.” and “Chuzhou, Anhui Prov.” This solves most the problems.

=> We have modified the text on the figure following your recommendation, omitting the abbreviation “prov.”. The text now reads “Hefei (Anhui)”.

Fig. 1. As proposed by our previous comment, some colors used in the figure were similar. The major problem raised from colors of D. suweonensis and D. flaviventris; these two are not easily distinguished by color-blinded people. Therefore, please change one of them.

==> We have modified the range of D flaviventris, it is now orange, and all figures with colour coding have been updated accordingly.

Fig. 2. I think you submitted the old version; please update. The photos, the sonograms, and the labels are still overlapping with the STRUCTURE probabilities.

=> The new figure is indeed very similar, only the height of some barplots and the values of the delta K changed following the additional analyses. The sonograms are now removed from the newly submitted figure, but we maintained labels and pictures on the graph as it is not overlapping with the sections of the bar plots that are showing multiple clade assignment. 

Materials and methods, lines 103 – 104:

I suspected that you have misunderstood the meaning of the reviewer. There was a missing link about how you identify, and how you assign your samples to these three clades. Therefore, the reviewer was asking about the definition of “the three clades”, not the definition of “what a clade means”. I think you need a few sentences to clarify how you distinguish, how you identify, and how you assign the samples to the three clades. By sample locations? Morphology? Or some other pre-tests?

=> The individuals were assigned based on sample location, until genetic analyses, upon which samples from the Korean Peninsula were re-assigned to either of the two clades based on the results of the genetic analyses. This information was added to the manuscript such as: “Call properties and morphological differences between clades were not known before this study, and therefore individual sampled in China were assigned to D. immaculatus, based on range, and individuals sampled in R Korea and DPR Korea were assigned to either D. suweonensis or the new clade based on the genetic analyses”.

Minor revisions

Line 89: lack of a full stop at the end of the sentence.

=> Corrected as suggested

Line 1242: “includes”

=> Corrected as suggested

---

## [Editor Report · Decision Letter 2]

26 May 2020

Yellow sea mediated segregation between North East Asian Dryophytes species

PONE-D-19-29311R2

Dear Dr. Borzée,

We are pleased to inform you that your manuscript has been judged scientifically suitable for publication and will be formally accepted for publication once it complies with all outstanding technical requirements.

With kind regards,

Si-Min Lin, Ph.D.

Academic Editor

PLOS ONE

Additional Editor Comments (optional):

Dear Dr. Borzée and Dr. Min,

Congratulation! I think the manuscript could be accepted in its current manner. Congratulations again for your good works!

Si-Min LIN
---

## [Editor Report · Acceptance letter]

11 Jun 2020

PONE-D-19-29311R2 

Yellow sea mediated segregation between North East Asian *Dryophytes* species 

Dear Dr. Borzée:

I'm pleased to inform you that your manuscript has been deemed suitable for publication in PLOS ONE. Congratulations! Your manuscript is now with our production department. 

Kind regards, 

on behalf of

Dr. Si-Min Lin 

Academic Editor

PLOS ONE